# The Critical Role of Volatile Organic Compounds Emission in Nitrate Formation in Lhasa, Tibetan Plateau: Insights from Oxygen Isotope Anomaly Measurements

Xueqin Zheng[a], Junwen Liu[a*], Nima Chuduo[b], Bian Ba[b], Pengfei Yu[a], Phu Drolgar[b], Fang Cao[c], Yanlin Zhang[c]

[a] College of Environment and Climate, Jinan University, Guangzhou, 511443, China

[b] Lhasa Meteorological Administration, Lhasa, 850010, China

[c] School of Ecology and Applied Meteorology, Nanjing University of Information Science and Technology, Nanjing 210044, China

[*] Corresponding author: Junwen Liu

Email: liu.junwen@jnu.edu.cn

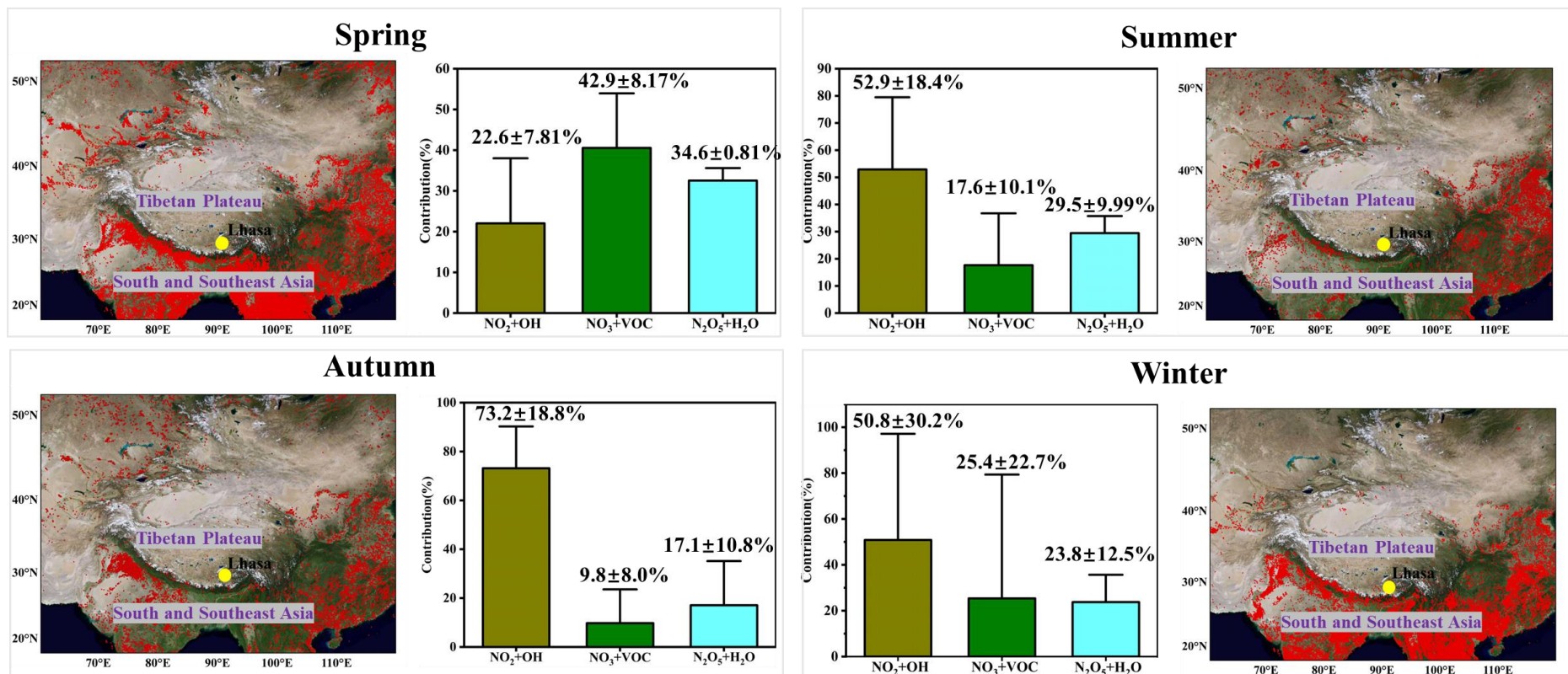

**TOC art**

## Abstract

Atmospheric particulate nitrate aerosol ($NO_3^-$), produced via the oxidation of nitrogen oxides ($NO_x$ = NO + $NO_2$), plays an important role in atmospheric chemistry and air quality, yet its formation mechanism remains poorly constrained in the plateau region. In this study, we reported for the first time the yearly variation in the signatures of the stable oxygen isotope anomaly ($\Delta^{17}O = \delta^{17}O - 0.52 \times \delta^{18}O$) in $NO_3^-$ collected in the urban Lhasa (3650 m a.s.l), on the Tibetan Plateau, China. Our results show that $NO_2$ + OH was the largest contributor to $NO_3^-$ formation (46 ± 26%), followed by $NO_3$ + VOC (26 ± 18%), and $N_2O_5$ + $H_2O$ (28 ± 11%) using the Bayesian Isotope Mixture Model. Notably, there were significant differences in the $NO_2$ + OH, $NO_3$ + VOC, and $N_2O_5$ + $H_2O$ pathways between spring and the other three seasons (T test, $p < 0.05$). By Hybrid Single-Particle Lagrangian Integrated Trajectory (HYSPLIT) dispersion model, we highlighted the influence of VOC emissions from regions such as Afghanistan and northern India, which enhanced $NO_3^-$ concentrations in Lhasa during spring. Furthermore, the diurnal distribution of $NO_3^-$ oxidation pathways varied distinctly across seasons, suggesting that these differences in $NO_3^-$ pathways are attributed to aerosol liquid water content (ALWC), volatile organic compounds (VOC) concentrations, and atmospheric lifetime of $NO_3^-$.

**Keywords:** nitrate, $\Delta^{17}O$-$NO_3^-$, oxidation pathways, Lhasa, VOC

## 1. Introduction

Nitrate aerosol ($NO_3^-$) is a key component regulating the mass concentration of atmospheric fine particulate matter ($PM_{2.5}$), which is highly related with air quality (Colmer et al., 2020), public health (Zhang et al., 2019; Zhang et al., 2017; Geng et al., 2021), and climate system (Clark and Tilman, 2008). Globally, the mass contribution of $NO_3^-$ in $PM_{2.5}$ is in the range of 5-30% (Huang et al., 2014; Xu et al., 2019; Salameh et al., 2015; Espina-Martin et al., 2024; Bell et al., 2007; Sun et al., 2022), depending on the locations and the severities of air pollution. For example, it was reported that $NO_3^-$ accounts for 22%, 27% and 26% of $PM_{2.5}$ in megacities in China (Zong et al., 2020), Europe (Espina-Martin et al., 2024) and U.S. (Sun et al., 2022), respectively. In addition, some studies found that the contribution of $NO_3^-$ would increase by 3-8 times with the occurrence of the particular-derived haze pollution (Ge et al., 2024; Song et al., 2019; Yin et al., 2022; Walters et al., 2024).

It is well-known that atmospheric $NO_3^-$ is formed by the oxidation of nitrogen oxides ($NO_x$=NO+$NO_2$) with different oxidants such as $O_3$, OH and $RO_2$ (Text S1). In general, atmospheric chemical transportation models are employed to depict the detailed oxidation pathways of $NO_3^-$ formation. However, there remains considerable uncertainty in modelling the contribution of individual oxidation pathways to $NO_3^-$ formation, particularly the $N_2O_5$ + $H_2O$ pathway, due to the wide variability of key parameters such as the $N_2O_5$ uptake coefficient, which has been shown to vary significantly depending on aerosol composition, relative humidity, and temperature. For example, it was reported that the predicted $N_2O_5$ uptake to $NO_3^-$ formation in Beijing, as estimated using WRF-Chem, ranges from 5% to 21% (Su et al., 2017). Higher contributions between 66% and 85% have been observed when applying the CMAQ model in Beijing (Qiu et al., 2019). Therefore, the application of alternative techniques is crucial for providing more reliable estimates and enhancing our understanding of $NO_3^-$ formation mechanisms, in addition to the insights gained from atmospheric chemical transportation models.

Stable oxygen isotope anomaly ($\Delta^{17}O = \delta^{17}O - 0.52 \times \delta^{18}O$) is recognized as a powerful tool to track formation pathways of atmospheric $NO_3^-$ (Zhang et al., 2024; Feng et al., 2023). This is because the oxygen atoms in the terminal positions of $O_3$ exhibit an elevated $\Delta^{17}O$ ($\Delta^{17}O = 39 \pm 2‰$) (Vicars and Savarino, 2014), whereas the $\Delta^{17}O$ values of other atmospheric oxidants (e.g., $H_2O$, OH, and $RO_2$) that can be incorporated to $NO_3^-$ are very close to 0‰. (Dubey et al., 1997; Barkan and Luz, 2003;

Alexander et al., 2020) Therefore, $\Delta^{17}O(NO_3^-)$ serves as a unique tracer of $O_3$ involvement in its formation pathways, offering valuable insights into the relative contributions of individual reactions. In recent years, the use of $\Delta^{17}O(NO_3^-)$ to elucidate $NO_3^-$ formation has garnered considerable attention. Walters et al. (2024) reported that the major formation pathways of annual $HNO_3$ production in the northeastern U.S. were $NO_2+OH$ (46%), $N_2O_5$ uptake (34%), and organic nitrate hydrolysis (12%), with notable seasonal variability. Additionally, Zhang et al. (2022) observed that the contribution of nocturnal chemistry to $NO_3^-$ formation increased at night, peaking at 72% around midnight. In contrast, the contribution of $NO_2+$ OH rose with sunrise, reaching its highest fraction (48%) around noon. However, nearly all current $\Delta^{17}O$-related observations have been conducted in the plain cities, with little attention given to plateau cities, where atmospheric conditions generally suffer from distinct energy consumption patterns and unique climatic factors (e.g., intense solar radiation). In this study, we present detailed results from comprehensive field observations conducted in Lhasa (3650 m a.s.l), one of the highest cities in the world, located on the Tibetan Plateau, China. For the first time, we quantify the relative contribution of three oxidation pathways to $NO_3^-$ formation in Lhasa on the basis of ambient measurements for $\Delta^{17}O$ signatures in $NO_3^-$.

## 2. Materials and methods

### 2.1 Sampling campaign

$PM_{2.5}$ samples were collected on the roof of a building (~15 m above ground) at the Meteorological Bureau of Lhasa (91.08°E, 29.40°N; Figure 1) in China. Lhasa, the capital of the Tibet Autonomous Region, is a rapidly developing city with a population of ~ 950000 and an urban area of ~ 30000 $km^2$ (Lhasa). The sampling site is surrounded by mixed land use, including residential areas, government offices, religious temples and commercial zones, with minimal heavy industry. The strong solar radiation and large diurnal temperature variations in this sampling site can lead to pronounced changes in boundary layer height, which in turn significantly influence vertical mixing and the transport of air pollutants.

The sampling campaign was conducted from June 2022 to July 2023 using a high-volume $PM_{2.5}$ sampler, which operated at a flow rate of 1.0 $m^3$/min. Samples were collected once a week, with each sampling session lasting 48 hours, except during intensive sampling periods in the summer (June 30

to July 14, 2022) and winter (January 28 to February 7, 2023). During these intensive periods, each sample was collected for 12 hours, from 8:00 to 20:00 and 20:00 to 8:00 on the following day, respectively. During the autumn of 2022, Lhasa experienced intermittent COVID-19 control measures, including restricted movement, reduced traffic activity, and temporary lockdowns in urban areas (Daily). Before sampling, all quartz filters (8 in. × 10 in., Pallflex) were calcined in a muffle furnace at 450 °C for 6 h to prevent impurities from contaminating the collected PM$_{2.5}$ samples. After sampling, the samples were collected and stored in a freezer at -20°C.

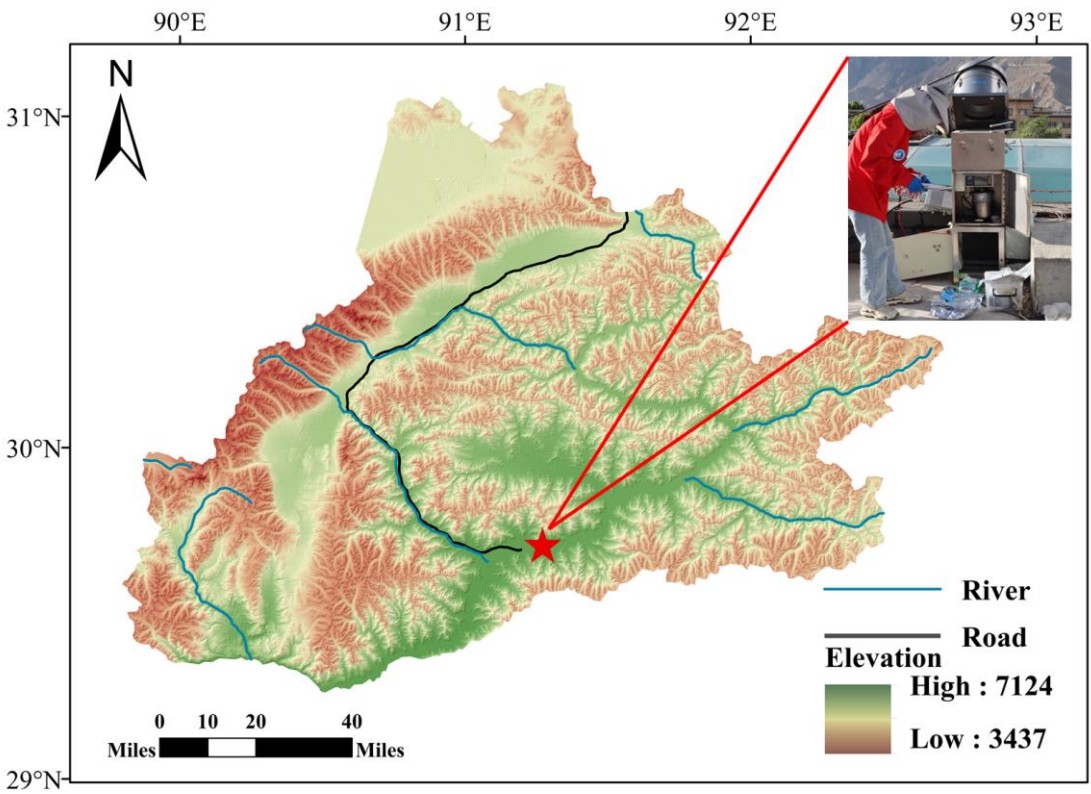

*Figure 1. Geographic position of sampling site in Lhasa, China.*

## 2.2 Measurements of water-soluble ions and isotopes

Water-soluble ions were measured by an ion chromatography (Dionex ICS-5000, Thermo Scientific Inc.) (Chen et al., 2022). In brief, a part of filter membranes (4.54 cm$^2$) was cut using a 17 mm diameter punch and placed in a 15 mL centrifuge tube with 10 mL of 18.2 MΩ ultrapure water. The tube was then subjected to ultrasonic treatment in an ice-water bath for 30 min to prevent ion volatilization. The extract was filtered through a 0.22 μm filter into a 30 mL sample bottle. This process was repeated with an additional 10 mL of water to ensure full extraction. The final extract was analyzed by an ion chromatography. The method detection limits (MDLs) for Cl$^-$, NO$_3^-$, SO$_4^{2-}$, Na$^+$, NH$_4^+$, K$^+$,

$Mg^{2+}$, and $Ca^{2+}$ were 0.001 mg/L, 0.001 mg/L, 0.003 mg/L, 0.02 mg/L, 0.01 mg/L, 0.02 mg/L, 0.006
mg/L, and 0.02 mg/L, respectively.
Stable oxygen isotopes ($\delta^{17}O$, $\delta^{18}O$, $\Delta^{17}O$, and $\Delta^{17}O = \delta^{17}O - 0.52 \times \delta^{18}O$) of $NO_3^-$ were
determined using an isotope ratio mass spectrometer (MAT253, Thermo Fisher Scientific, USA) at
Nanjing University of Information Science and Technology (Fan et al., 2021; Zhang et al., 2022).
Briefly, $NO_3^-$ from filter extractions (containing at least 0.8 μg N) were converted into gaseous $N_2O$
using the bacterial denitrifier method. $N_2O$ was then further thermally decomposed into $O_2$ and $N_2$ in
a gold tube heated to 800°C. The produced $O_2$ was analyzed for oxygen isotopes by isotope ratio mass
spectrometer. The duplicated analysis showed that the errors were within 1.32% for $\Delta^{17}O$-$NO_3^-$.
**2.3 Primary data sources**
Meteorological parameters, including ambient temperature (T), relative humidity (RH), rainfall,
radiation, wind direction (WD) and wind speed (WS) during the sampling campaign, were obtained
from the Meteorological Bureau of Lhasa. Additionally, $NO_2$ and $O_3$ during the sampling campaign
were downloaded from the National Meteorological Information Center (https://air.cnemc.cn:18007/).
**2.4 Evaluation of $NO_3^-$ oxidation pathways**
In our study, we aimed to quantify the relative contribution of different oxidation pathways to
$NO_3^-$ production based on $\Delta^{17}O$-$NO_3^-$. Due to the low $Cl^-$ concentrations observed in Lhasa, the $NO_3^-$
formation pathways considered in this study are limited to $NO_2 + OH$, $NO_3 + VOC$, and $N_2O_5 + H_2O$.
Although $NO_3 + VOC$ was generally considered a minor pathway in continental regions (Alexander et
al., 2009), we included it because elevated VOC concentrations were observed at our sampling site in
Lhasa, influenced by both biogenic emissions (e.g. incense burning) and anthropogenic sources (e.g.
vehicle emissions) (Tang et al., 2022). The relative contributions of the three pathways were
determined using a $\Delta^{17}O$-based mass balance approach (Michalski et al., 2003), as shown in Equations
(1) and (2):
$$\Delta^{17}O\text{-}NO_3^- = (\Delta^{17}O\text{-}NO_3^-)_{NO2+OH} \times f_{NO2+OH} + (\Delta^{17}O\text{-}NO_3^-)_{NO3+VOC} \times f_{NO3+VOC}$$
$$+ (\Delta^{17}O\text{-}NO_3^-)_{N2O5+H2O} \times f_{N2O5+H2O} \quad (1)$$
$$f_{NO2+OH} + f_{NO3+VOC} + f_{N2O5+H2O} = 1 \quad (2)$$
where $\Delta^{17}O$-$NO_3^-$ value is the $\Delta^{17}O$ value of $NO_3^-$ in $PM_{2.5}$. The $(\Delta^{17}O\text{-}NO_3^-)_{NO2+OH}$, $(\Delta^{17}O\text{-}$
$NO_3^-)_{NO3+VOC}$, and $(\Delta^{17}O\text{-}NO_3^-)_{N2O5+H2O}$ correspond to the $\Delta^{17}O$ values from $NO_2+OH$, $NO_3+VOC$ and

N$_2$O$_5$+H$_2$O, respectively. The $\Delta^{17}$O values for each pathway were calculated using Equations (3), (4), and (5) (Savarino et al., 2016; Alexander et al., 2009):

$$(\Delta^{17}\text{O-NO}_3^-)_{\text{NO2+OH}} \text{ (‰)}=2/3\alpha\times\Delta^{17}\text{O-O}_3\text{* (3)}$$

$$(\Delta^{17}\text{O-NO}_3^-)_{\text{NO3+VOC}} \text{ (‰)}=2/3\alpha\times\Delta^{17}\text{O-O}_3\text{*}+1/3\times\Delta^{17}\text{O-O}_3\text{* (4)}$$

$$(\Delta^{17}\text{O-NO}_3^-)_{\text{N2O5+H2O}} \text{ (‰)}=1/3\alpha\times\Delta^{17}\text{O-O}_3\text{*}+1/2(2/3\alpha\times\Delta^{17}\text{O-O}_3\text{*}+1/3\times\Delta^{17}\text{O-O}_3\text{*}) \text{ (5)}$$

Previous studies have demonstrated a linear correlation between $\Delta^{17}$O-O$_3$ and $\Delta^{17}$O-O$_3$*, with $\Delta^{17}$O(O$_3$) values ranging from 20% to 40% in tropospheric O$_3$ (Vicars and Savarino, 2014; Ishino et al., 2017). The equations are shown as follows (Vicars et al., 2012):

$$\Delta^{17}\text{O-O}_3\text{*}=1.5\times\Delta^{17}\text{O-O}_3 \text{ (6)}$$

Based on previous observations of tropospheric O$_3$, $\Delta^{17}$O-O$_3$* average value was approximately 39‰. The $\alpha$ value represents the proportional contribution of O$_3$ to the NO oxidation pathway and can be estimated using the following equations (7) (Alexander et al., 2009). When NO$x$ is in photochemical steady state, $\Delta^{17}$O-NO$_2$ can be represented using the following equation (10):

$$\alpha=K_{P1} [\text{O}_3] \times [\text{NO}]/(K_{P1} \times [\text{O}_3] \times [\text{NO}]+K_{P2} \times [\text{NO}] \times [\text{HO}_2]+ K_{P3} \times [\text{NO}] \times [\text{RO}_2]) \text{ (7)}$$

$$K_{P1}=3.0\times10^{-12}\times e^{(-1500/T)} \text{ (8)}$$

$$K_{P2}=K_{P3}=3.5\times10^{-12}\times e^{(270/T)}(\text{cm}^3\cdot\text{molecule}^{-1}\cdot\text{s}^{-1}) \text{ (9)}$$

$$\Delta^{17}\text{O-NO}_2=\alpha\Delta^{17}\text{O-O}_3\text{* (10)}$$

where T represents the ambient temperature (K) (Kunasek et al., 2008). The HO$_2$ mixing ratios were estimated using empirical equations in the absence of direct HO$_2$ observations (Kanaya et al., 2007). Due to the lower temperatures in Lhasa during non-summer seasons, HO$_2$ concentrations were assessed using a formula derived from winter conditions.

Winter

$$[\text{HO}_2\cdot]/\text{ppt} = \exp (5.7747 \times 10^{-2} [\text{O}_3] \text{ (ppb)} – 1.7227) \text{ for daytime (11)}$$

$$[\text{HO}_2\cdot]/\text{ppt} = \exp (7.7234 \times 10^{-2} [\text{O}_3] \text{ (ppb)} – 1.6363) \text{ for nighttime (12)}$$

Summer

$$[\text{HO}_2\cdot]/\text{pptv} = \exp (2.0706 \times 10^{-2} [\text{O}_3] \text{ (ppb)} + 1.0625) \text{ for daytime (13)}$$

$$[\text{HO}_2\cdot]/\text{pptv} = 0.2456 + 0.1841 [\text{O}_3] \text{ (ppb) for nighttime (14)}$$

**2.5 Stable isotope analysis in the R (SIAR) model**

In this study, stable isotope analysis in the R (SIAR) model was employed to estimate the relative contributions of three main pathways to NO$_3^-$ (Parnell et al., 2010). The SIAR model is well-suited for

analyzing multiple formation pathways, as it effectively incorporates uncertainties and parameter variability, leading to more reliable estimates. Specifically, this model allows for a detailed analysis of oxygen isotope ($\Delta^{17}O$), enabling accurate modeling of $NO_3^-$ formation pathways based on oxygen isotope measurements. The SIAR model is a Bayesian mixture model, mathematically formulated as follows:

$$X_i = \sum_{j=1}^{K} p_j \times f_{ij}$$

$$p_1 + p_2 + \cdots + p_k = 1$$

$$f_{ij} \sim N\left(\mu_j, \omega_j^2\right)$$

Where $X_i$ is the observed $\Delta^{17}O$ values for sample $i$ ($i = 1, 2, 3, ..., N$) and $p_j$ is the proportional contribution of each $NO_3^-$ formation pathway $j$ to the sample $i$. $f_{ij}$ is the $\Delta^{17}O$ values of formation pathway $j$ for sample $i$ and follows a normal distribution with mean ($\mu_j$) and variance ($\omega_j^2$). Within the Bayesian framework, prior distributions are assigned to each $p_j$, and these are updated with the observed data $X_i$ to obtain posterior distributions, allowing for inference of the proportional contributions $p_j$ of each pathway.

**2.6 Aerosol liquid water content (ALWC) and the Hybrid Single-Particle Lagrangian Integrated Trajectory (HYSPLIT)**

To evaluate the influence of ALWC on the $NO_3^-$ formation, ALWC was calculated using the ISORROPIA II model developed by Fountoukis and Nenes (Fountoukis and Nenes, 2007). The ISORROPIA II model includes two modes: the forward mode, which requires the concentrations of both particulate and gaseous pollutants concentrations as inputs, and the reverse mode, which only requires the concentrations of particulate pollutants concentrations. The model computes the ALWC in both modes based on particulate pollutant concentrations (e.g., $NH_4^+$, $Na^+$, $Ca^{2+}$, $K^+$ and $Mg^{2+}$), as well as ambient RH and T. In this study, the reverse mode was employed due to the lack of gaseous pollutant concentrations observations.

Additionally, the Hybrid Single-Particle Lagrangian Integrated Trajectory (HYSPLIT) model was utilized to compute 72-hour back trajectories during the sampling campaign. HYSPLIT, developed by the National Oceanic and Atmospheric Administration Air Resources Laboratory (NOAA/ARL), is available on their website (https://www.ready.noaa.gov/HYSPLIT.php). This model has been widely

used for simulating the transport and dispersion trajectories of pollutants such as $PM_{2.5}$, VOC, $O_3$, and
$NO_x$, among others (He et al., 2022; Zhao et al., 2015; Cao et al., 2023). Backward trajectories for each
sampling day were calculated at an altitude of 3650 meters using meteorological data from the Global
Data Assimilation System (GDAS), available through the US Air Resources Laboratory (NOAA ARL)
(https://www.ready.noaa.gov/data/archives/gdas1/).

## 3. Results

### 3.1 Overview of the meteorological parameters in Lhasa during the sampling campaign

Figure 2a presents the daily variations in meteorological parameters, including temperature,
relative humidity (RH), rainfall and solar radiation. During the sampling campaign, the annual average
temperature was 11.5°C, ranging from -2.83 to 24.2°C. The highest average temperature was observed
in summer (19.7°C), while the lowest (3.11°C) was recorded in winter. Relative humidity (RH) varied
between 6.67 and 66.8%, with the lowest average RH occurring in winter (17.1%) and the highest in
summer (35.6%). The near-surface layer of Lhasa is influenced by a thermal low-pressure system, and
the southwest monsoon, active between June and September, transports moisture-laden air from the
Indian Ocean, resulting in increased rainfall during summer. Solar radiation intensity exhibited a
seasonal trend consistent to those of temperature and RH, peaking in summer (394 $W/m^2$) and reaching
its lowest levels in winter (220 $W/m^2$). The dominant wind direct (WD) was southeast in spring, but
southwest in the other three seasons (Figure 3). Wind speed (WS) was highest in spring but lowest in
autumn.

### 3.2 $NO_3^-$ concentration

$NO_3^-$ mass concentrations ranged from 0.10 to 1.72 $\mu g/m^3$, with an average value of $0.62 \pm 0.31$
$\mu g/m^3$. $NO_3^-$ concentrations exhibited distinct seasonal patterns. As shown in Figure S1, the equivalent
concentrations of $[SO_4^{2-} + NO_3^-]$ were considerably higher than those of $[NH_4^+]$, indicating that $NH_4^+$
was insufficient to fully neutralize $NO_3^-$. This suggests that a portion of $NO_3^-$ may have existed in other
forms, such as $KNO_3$ and $Ca(NO_3)_2$. This inference is supported by the strong positive correlations
between $NO_3^-$ and $K^+$ ($r = 0.64$, $p < 0.1$) and $Ca^{2+}$ ($r = 0.43$, $p < 0.01$), especially in spring, as shown
in Figure S2. In contrast, $NO_3^-$ showed relatively weak negative correlations with T ($r = -0.27$, $p < 0.01$)
and RH ($r = -0.22$, $p < 0.1$), indicating that under the specific atmospheric conditions in Lhasa,
meteorological parameters might not be the dominant factors controlling the gas-particle partitioning
of $NO_3^-$. The maximum monthly average values of $NO_3^-$ concentration occurred in spring ($0.83 \pm 0.35$
$\mu g/m^3$) with the instantaneous maximum reaching 1.72 $\mu g/m^3$, whereas the lowest was recorded in
autumn ($0.23 \pm 0.13$ $\mu g/m^3$) with an instantaneous minimum of only 0.09 $\mu g/m^3$ (Table 1). The elevated
$NO_3^-$ concentrations in spring could be attributed to biomass burning emitted from south and Southeast
Asia (Figure S3/Figure S4). The strong between $NO_3^-$ and $K^+$ in spring further this explanation.

247         In spring, high $NO_3^-$ concentrations were associated with weak southeasterly winds (< 3 m/s) in

the bivariate polar plot, suggesting probable impacts from local emissions (Figure 3). The southeasterly
sector of sampling site includes residential areas, agriculture land and major transportation routes,
which are potential NO$x$ sources. In spring, intensified agriculture activities (e.g., fertilization, biomass
burning) might increase NO$x$ emissions. Meanwhile, low wind speeds likely limit atmospheric
dispersion, promoting the local accumulation of precursors and enhancing $NO_3^-$ production. During
the rainy summer, shorter $NO_3^-$ lifetimes indicated a weak influence from regional transport, with a
more pronounced contribution from local emissions. In autumn, $NO_3^-$ concentrations were relatively
low, which coincided with strict local COVID-19 restrictions in Lhasa. These measures significantly
reduced human activity and traffic, leading to suppressed local emissions. Despite low wind speeds
typically favor pollutant accumulation, $NO_3^-$ concentrations remained low, suggesting that both
reduced local sources and seasonal meteorological conditions constrained $NO_3^-$ production.
Nevertheless, the persistence of measurable $NO_3^-$ under such stagnant conditions also implied a
potential contribution from regional transport during this period. In winter, elevated $NO_3^-$
concentrations under low wind speeds (< 3 m/s) emphasized the significant contribution of local
emissions. These findings underscored that both regional transport and local emissions were important
contributors to $NO_3^-$ concentrations in Lhasa. Furthermore, based on our day-night sampling scheme,
no nychthemeral (day-night) differences in $NO_3^-$ concentrations were detected (Table S1). A similar
day-night pattern of $NO_3^-$ concentrations also has been observed in in Beijing (Luo et al., 2020).
Table 1 Average values of water-soluble ions and $\Delta^{17}O\text{-}NO_3^-$ during the sampling campaign

| | | $Na^+$ $\mu g/m^3$ | $NH_4^+$ $\mu g/m^3$ | $K^+$ $\mu g/m^3$ | $Mg^{2+}$ $\mu g/m^3$ | $Ca^{2+}$ $\mu g/m^3$ | $Cl^-$ $\mu g/m^3$ | $NO_3^-$ $\mu g/m^3$ | $SO_4^{2-}$ $\mu g/m^3$ | $\Delta^{17}O\text{-}NO_3^-$ ‰ |
|---|---|---|---|---|---|---|---|---|---|---|
| Annual | Minmum | 0.02 | 0 | 0.004 | 0.004 | 0.004 | 0.004 | 0.09 | 0.06 | 18.3 |
| | Maximum | 0.68 | 1.22 | 0.29 | 0.08 | 3.52 | 0.51 | 1.72 | 2.37 | 34.1 |
| | Average | 0.16 | 0.3 | 0.07 | 0.02 | 1.09 | 0.08 | 0.62 | 0.74 | 26.3 |
| | Std.Dev | 0.14 | 0.26 | 0.06 | 0.01 | 0.7 | 0.1 | 0.31 | 0.45 | 3.13 |
| spring | Minmum | 0.04 | 0.16 | 0.04 | 0.01 | 1.02 | 0.01 | 0.45 | 0.6 | 27.2 |
| | Maximum | 0.16 | 1.22 | 0.2 | 0.05 | 2.56 | 0.05 | 1.72 | 2.14 | 30.4 |
| | Average | 0.09 | 0.52 | 0.09 | 0.02 | 1.67 | 0.03 | 0.83 | 1.11 | 28.8 |
| | Std.Dev | 0.03 | 0.3 | 0.04 | 0.01 | 0.51 | 0.01 | 0.35 | 0.52 | 0.99 |
| summer | Minmum | 0.02 | 0 | 0.01 | 0.01 | 0.03 | 0.003 | 0.13 | 0.18 | 20.2 |
| | Maximum | 0.4 | 1.08 | 0.09 | 0.04 | 2.4 | 0.13 | 1 | 2.37 | 28.5 |
| | Average | 0.09 | 0.18 | 0.03 | 0.02 | 1.15 | 0.3 | 0.5 | 0.72 | 25.5 |
| | Std.Dev | 0.08 | 0.17 | 0.02 | 0.01 | 0.5 | 0.3 | 0.23 | 0.45 | 2.2 |
| autumn | Minmum | 0.02 | 0.003 | 0.004 | 0.01 | 0.004 | 0.01 | 0.09 | 0.06 | 21.2 |
| | Maximum | 0.17 | 0.11 | 0.1 | 0.03 | 0.24 | 0.17 | 0.51 | 0.55 | 24.9 |
| | Average | 0.09 | 0.04 | 0.3 | 0.02 | 0.13 | 0.05 | 0.23 | 0.31 | 23.05 |
| | Std.Dev | 0.05 | 0.04 | 0.3 | 0.01 | 0.08 | 0.05 | 0.13 | 0.14 | 1.44 |
| winter | Minmum | 0.06 | 0.09 | 0.02 | 0.01 | 0.05 | 0.04 | 0.21 | 0.32 | 18.3 |
| | Maximum | 0.56 | 0.87 | 0.29 | 0.08 | 3.52 | 0.51 | 1.46 | 1.57 | 34.1 |
| | Average | 0.19 | 0.44 | 0.12 | 0.03 | 1.04 | 0.16 | 0.75 | 0.73 | 25.9 |
| | Std.Dev | 0.12 | 0.21 | 0.08 | 0.02 | 0.78 | 0.13 | 0.28 | 0.34 | 3.86 |




**3.3 Oxygen isotopes of $NO_3^-$**

To explore the three major oxidation pathways of $NO_3^-$ formation, 53 samples representing varying $NO_3^-$ concentrations across different seasons were selected for oxygen isotope measurements (Figure 2b). The $\Delta^{17}O\text{-}NO_3^-$ values ranged from 18.3 to 34.1‰, with an average of 26.3 ± 3.13‰, which is slightly lower than the global average of 28.6 ± 4.5‰ simulated by the Global Chemical Transport Model (Alexander et al., 2020). As shown in Table S2, the observed $\Delta^{17}O\text{-}NO_3^-$ values in this study were similar to most mid- and low-latitude regions, but lower than those in polar regions (~ 32‰). As listed in Table S1, the average $\Delta^{17}O\text{-}NO_3^-$ values in spring, summer, autumn and winter were 28.8 ± 8.0‰, 25.5 ± 2.20‰, 25.6 ± 1.35‰ and 25.9 ± 3.56‰, respectively. The differences in $\Delta^{17}O\text{-}NO_3^-$ values between spring and summer, as well as between spring and winter, were statistically significant ($p < 0.05$). The elevated $\Delta^{17}O\text{-}NO_3^-$ values in spring could be attributed to a higher proportion of nocturnal pathways that enrich $\Delta^{17}O\text{-}NO_3^-$ values, such as $NO_3$ + VOC and $N_2O_5$ + $H_2O$ pathway. In contrast, the lower $\Delta^{17}O\text{-}NO_3^-$ values in other three seasons suggested a greater production of $NO_3^-$ formation via $NO_2$ + OH pathway, leading to more negative $\Delta^{17}O\text{-}NO_3^-$ values. Diurnal variation in $\Delta^{17}O\text{-}NO_3^-$ values also differed across season (Figure S5). In summer, the average of $\Delta^{17}O\text{-}NO_3^-$ values during the day (25.3 ± 2.39‰) was lower than at night (26.7 ± 1.03‰). Conversely, in winter, the average of $\Delta^{17}O\text{-}NO_3^-$ values during the day (28.0 ± 3.79‰) was significantly higher than at night (24.4 ± 3.85‰). Similar diurnal patterns, with higher daytime $\Delta^{17}O\text{-}NO_3^-$ values and lower nighttime values, have also been observed in winter in the U.S. (Vicars et al., 2013) and other cities in China (He et al., 2018).

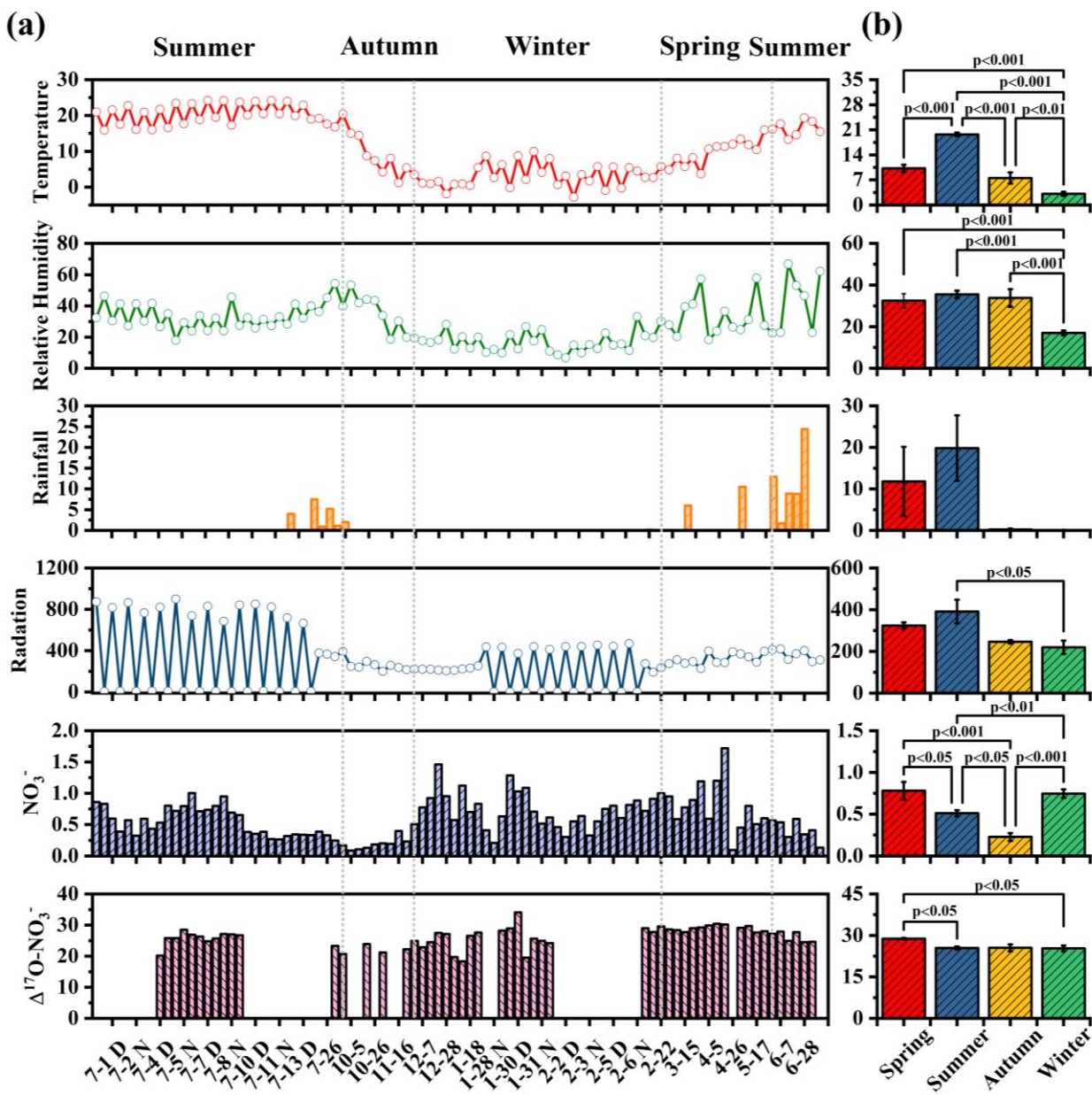


*Figure 2. (a) shows the time series of temperature (℃), relative humidity (%), Rainfall (mm), Radiation*
*(W/m², NO₃⁻ concentration (μg/m³), and Δ¹⁷O- NO₃⁻ (‰) from June  2022 to July 2023. (b) shows*
*the average values significance at different seasons with their statistical.*



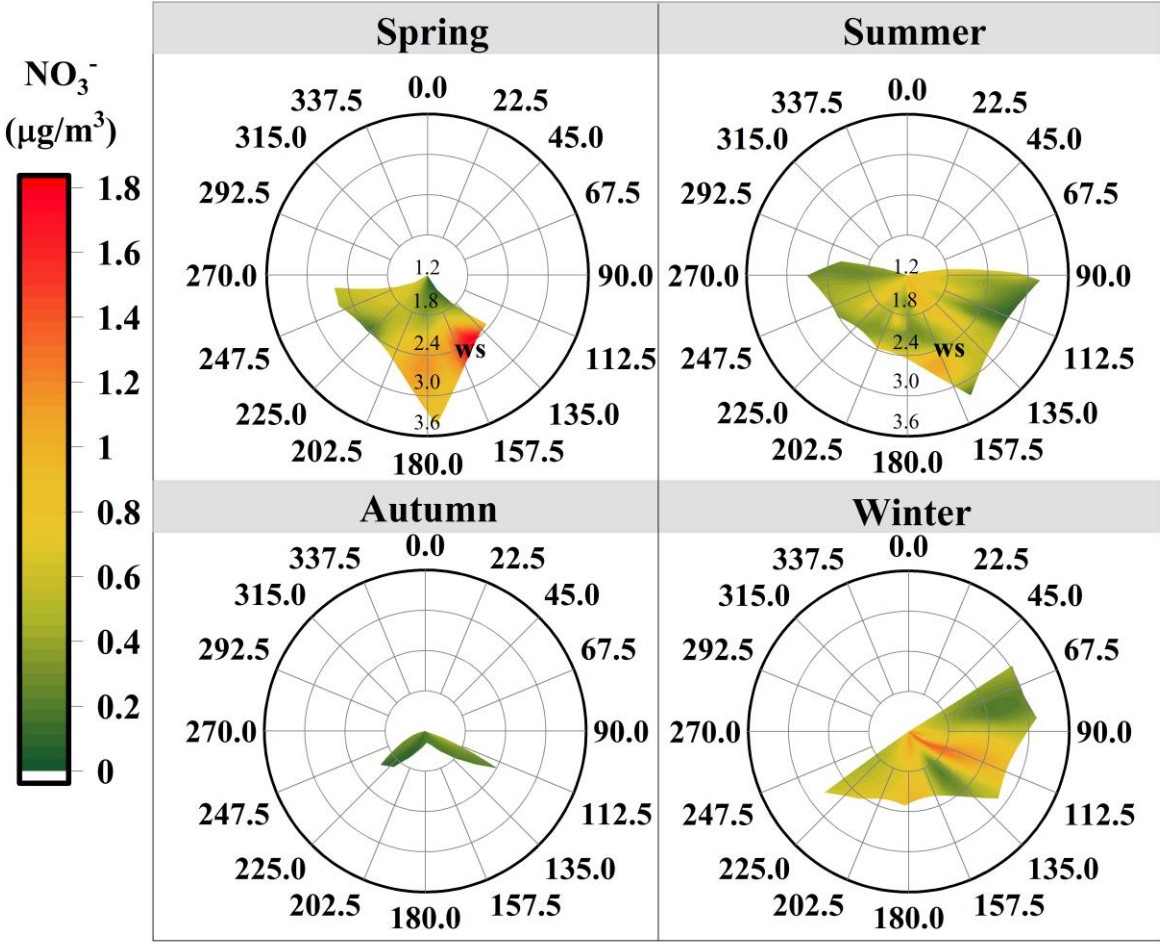


*Figure 3. Bivariate polar plot illustrates the seasonal variation in the mass concentration of NO₃⁻ in relation to wind speed (WS, m/s) and wind direction (WD, degrees).*


## 4. Discussion

### 4.1 A comparison of NO₃⁻ oxidation pathways in Lhasa with other megacities in plain regions

Typically, observations of $\Delta^{17}$O-NO₃⁻ and estimated α (the proportion of O₃ oxidation in NO₂ production rate) values are employed to quantify the contributions of major NO₃⁻ oxidation pathway in conjunction with a Bayesian model. The α value ranged from 0.63 to 0.93, with an average of 0.83 ± 0.06, suggesting the significance of O₃ participation in NO oxidation during the sampling campaign. On the other hand, our α values were lower than those (0.85-1) for other midlatitude regions (Alexander et al., 2009). The α values are influenced by the relative amount of O₃, HO₂ and RO₂ in NO$x$ cycling. Due to the generally high O₃ concentrations (O₃ > 50 ppb) observed in Lhasa, nearly all

α values exceeded 0.8 (Figure S6). To evaluate the impact of key parameters on the estimated
contributions of different $NO_3^-$ formation pathways, we conducted a sensitivity analysis by assumed
the α values and $\Delta^{17}O$ value of the terminal oxygen atoms of $O_3$ ($\Delta^{17}O\text{-}O_3^*$). As listed in Table S3, the
assumption of α and $\Delta^{17}O\text{-}O_3^*$ have an impact on the $NO_3^-$ formation mechanisms. When $\Delta^{17}O\text{-}O_3^*$
was fixed at 39‰, increasing α from 0.7 to 0.9 led to a notable increase in the relative contribution of
the $NO_2$ + OH pathway from 25% to 46%, while that of the $NO_3$ + VOC pathway decreased from 46%
to 25%. The $N_2O_5$ + $H_2O$ pathway remained nearly constant, with contributions ranging between 28%
and 29%, indicating that this pathway is relatively insensitive to changes in α values. Similarly, when
α was varied within a reasonable range (0.68-0.93), increasing the $\Delta^{17}O\text{-}O_3^*$ value from 37‰ to 39‰
led to an increase in the $NO_2$ + OH contribution from 37% to 46%, and a corresponding decrease in
the $NO_3$ + VOC contribution from 35% to 26%. Again, the $N_2O_5$ + $H_2O$ contribution remained stable
at ∼ 28%. These results suggest that the estimated contributions of $NO_2$ + OH and $NO_3$ + VOC
pathways are sensitive to assumptions about α and $\Delta^{17}O\text{-}O_3^*$, whereas the contribution of the $N_2O_5$ +
$H_2O$ pathway is relatively robust under the tested conditions. Because Lhasa is characterized by
relatively high VOC concentrations and $\Delta^{17}O\text{-}O_3^*$ is generally close to 39‰, we consider our
parameter assumptions reasonable for further estimating $NO_3^-$ formation pathways for each sample..
On average, the relative contributions of $NO_2$ + OH ($f_{NO2+OH}$), $NO_3$ + VOC ($f_{NO3+VOC}$) and $N_2O_5$
+ $H_2O$ ($f_{N2O5+H2O}$) to $NO_3^-$ formation in Lhasa during the sampling campaign were $46 \pm 26\%$, $26 \pm 19\%$
and $28 \pm 11\%$, respectively. To better understand the characteristics of $NO_3^-$ formation mechanism in
Lhasa, we performed a detailed comparison around the China for the relative contributions of key
oxidation pathways using the $\Delta^{17}O$ methodology (Figure 4). Overall, similar to most Chinese cities,
$NO_3^-$ formation in Lhasa was predominantly driven by the $NO_2$ + OH pathway, exhibiting distinct
seasonal and regional variations. In particular, the average $f_{NO3+VOC}$ values were generally several times
higher in spring in Lhasa than in other urban cities. Compared to rural/remote areas, the average
$f_{NO3+VOC}$ values showed higher fractions in Lhasa, revealing the influence of anthropogenic emission,
i.e., vehicle exhaust and heating, on $NO_3^-$ formation. In Lhasa, the Capital of Tibet, field measurements
among different years showed a substantial increase in VOC concentrations in urban areas of the Tibet
Plateau, comparable to those in North China (Tang et al., 2022), revealing the importance of the active
$NO_3$ + VOC pathway for $NO_3^-$ pollution formation in Lhasa. In fact, recent studies have recognized
$NO_3$ + VOC as a major formation mechanism for $NO_3^-$ production. For instance, Fan et al. (2021)

found that $f_{NO3+VOC}$ in Beijing increased from 17% in summer to 32% in winter based on $\Delta^{17}O$-$NO_3^-$ measurements. . He et al. (2018) estimated the relative contributions of $NO_3$ + VOC and $N_2O_5$ + $Cl^-$ to $NO_3^-$ formation and found that $NO_3$ + VOC and $N_2O_5$ + $Cl^-$ were in the range of 16-56%, underscoring the significant roles of these pathways during haze events in Beijing. Similarly, Feng et al. (2023) also reported that the $f_{NO3+VOC}$ values were up to 49.6% in winter in northern China. In Guangzhou,, Wang et al. (2023) noted that the average $f_{NO3+VOC}$ value was at the 488m (25%) higher than that at the ground (12%). Furthermore, Li et al. (2022) reported that $f_{NO3+VOC}$ increased from 5% in urban to 13.5% in rural regions in Northeast China. Although the specific nighttime $RO_2$ production mechanism in Lhasa remains unclear, studies in other cities have demonstrated that $NO_3$+VOC pathway was the dominant channel for nighttime $RO_2$ (Fisher et al., 2016), which in turn leads to the formation of alkyl and multifunctional nitrates ($RONO_2$) and eventually $NO_3^-$. In such cases, the $RO_2$ concentration is expected to be correlated with $NO_3$ radical production, which depends on the reaction rate of $O_3$ and $NO_2$ (Brown and Stutz, 2012). Given the relatively high nighttime $O_3$ concentrations in Lhasa, it is plausible that $O_3$-driven nighttime $NO_3$ chemistry plays an important role, thereby enhancing $NO_3$+VOC derived from $RO_2$ production and $NO_3^-$ formation. Global modelling studies also support the significant of this pathway. For instance, Alexander et al. (2020) reported that the $NO_3$ + VOC pathway via the $RONO_2$ mechanism accounted for 3% of global $NO_3^-$ formation on average. The relatively high $f_{NO3+VOC}$ values observed in Lhasa are broadly consistent with these findings, especially under conditions of high VOC concentrations and strong nighttime oxidant levels.

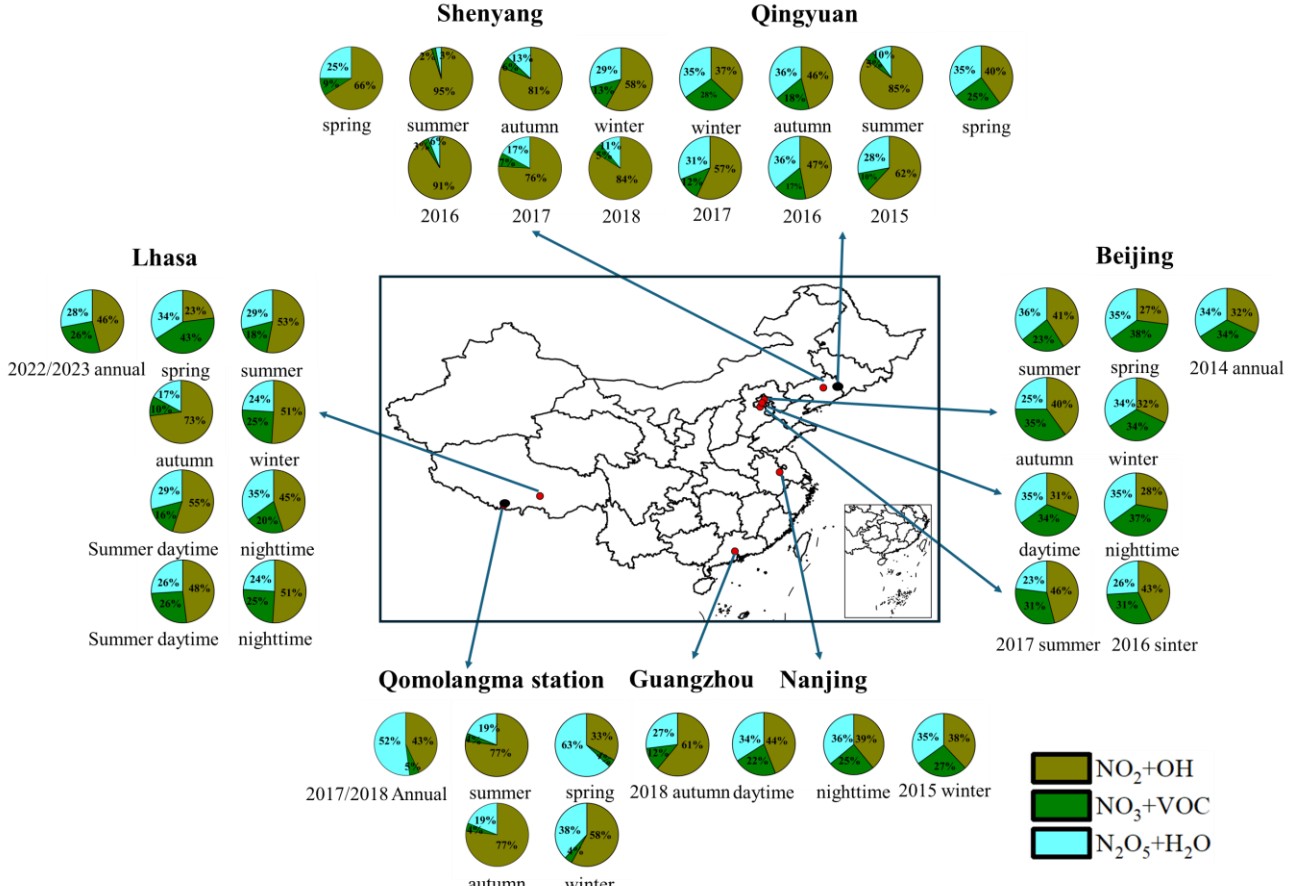

361

*Figure 4. Summary of the relative contributions of key oxidation pathways using the $\Delta^{17}O$ methodology around the China (data given in Table S4 in the Supplement). Colors for the study labels indicate the type of sampling location: urban areas (red), and rural/remote areas (black). The pie charts show the relative contribution of different pathways to $NO_3^-$ formation: $f_{NO2+OH}$ ( deep yellow), $f_{NO3+VOC}$ deep green), and $f_{N2O5+H2O}$ (light blue).*

### 4.2 Seasonal and diurnal variations of $NO_3^-$ oxidation pathways

Figure S7 illustrates the seasonal variations in the relative contributions of the three main oxidation pathways to $NO_3^-$ formation. When comparing different seasons, the $f_{NO2+OH}$ values were lower (p < 0.01) in spring (22.6%) than in winter (50.8%), summer (52.9%) and autumn (73.2%). The dominance of $NO_2$ + OH pathway in autumn is consistent with observations at Mt. Everest during the autumn seasons of 2017 and 2018, suggesting that $NO_3^-$ formation on the Tibetan Plateau in autumn may be mainly driven by $NO_2$ + OH pathway (Lin et al., 2021; Wang et al., 2020b).

A significant increase in the $f_{NO3+VOC}$ values was observed in spring (p < 0.05). First, $O_3$ and $NO_2$ are precursors of $NO_3$. In this work, the highest concentrations of $O_3$ were found in spring (114.9 ± 18.1 μg/m³), likely leading to elevated $NO_3$ concentrations. Additionally, the low temperature and

reduced OH radical concentrations in spring facilitate the reaction of $NO_2$ and $O_3$ to synthesize $NO_3$. This might be an appropriate reason for the $f_{NO3+VOC}$ values in spring. High-altitude locations such as Nepal (5079 m a.s.l.) and Qomolangma Station (4300 m a.s.l.) have experienced stratospheric ozone intrusions, especially in spring and winter, as reported in previous studies (Zhang et al., 2025; Cristofanelli et al., 2010; Morin et al., 2007; Zhang et al., 2022; Lin et al., 2016; Yin et al., 2017; Wang et al., 2020b). Notably, such intrusions in spring may elevate tropospheric $O_3$ levels in Lhasa, resulting in a mixture of tropospheric and stratospheric $O_3$ that enhances $NO_3^-$ production. Second, previous study has indicated that the Afghanistan-Pakistan-Tajikistan region, the Indo-Gangetic Plain, and Meghalaya-Myanmar region could transport industrial VOC to various zones in Tibet from west to east. Additionally, agricultural areas in northern India could contribute biomass burning-related VOC to the middle-northern and eastern regions of Tibet (Li et al., 2017). During our sampling campaign, South and Southeast Asia air clusters were notably prevalent in the springtime, coinciding with intensive fire spots observed in Afghanistan, Pakistan, India, Nepal, and Bhutan (Figure S3/S4). These observations, combined with the prevailing South and Southeast Asia air mass trajectories in spring, strongly suggest that long-range transported VOC from South Asia were delivered to Lhasa and likely participated in local $NO_3^-$ production via $NO_3$ + VOC pathway. Moreover, recent studies have shown that ambient VOC concentrations in the urban areas on the Qinghai-Tibet Plateau were comparable to those in the North China Plain (Tang et al., 2022). The input of VOC through long-range transport might further elevate VOC concentrations, thereby promoting $NO_3^-$ formation via $NO_3$ + VOC pathway and contributing to the enhanced $f_{NO3+VOC}$ values observed in spring. While VOC appears to play a dominant role in the process, it should be noted that other nitrogen species (e.g., NO, $NO_2$) associated with biomass burning emissions may also be transported over long distances and influence $NO_3^-$ formation in Lhasa. These co-transported nitrogen compounds, although not directly quantified in this study, could further contribute to $NO_3^-$ production in spring. Taken together, these findings provide strong evidence that long-range transport of biomass burning emissions, particularly from South Asia, can substantially influence springtime $NO_3^-$ formation in Lhasa.

Similarly, the $f_{N2O5+H2O}$ values exhibited its highest contributions during spring, with significant seasonal differences ($p < 0.05$) except when compared to summer ($p > 0.05$). Typically, high RH enhances $NO_3^-$ formation via $N_2O_5$ + $H_2O$ pathway. However, studies have revealed that during sandstorm events, a significant large $N_2O_5$ uptake coefficient is observed on urban aerosols in spring

408 (Xia et al., 2019). In this study, the mean $Ca^{2+}$ concentration in $PM_{2.5}$ was found to be the highest in

409 spring, suggesting a possible role of dust in facilitating $N_2O_5$ uptake. Additionally, $N_2O_5 + H_2O$

410 pathway has been reported to be promoted by elevated $NO_3^-$ concentrations (Lin et al., 2021), which

411 were also highest in spring. Therefore, the increased $f_{N2O5+H2O}$ values during spring might be attributed

412 to the combined effects of lower RH, elevated $Ca^{2+}$ levels, and high $NO_3^-$ concentrations.

413  Interestingly, distinct diurnal patterns of $NO_3^-$ oxidation pathways were observed during the

414 sampling campaign (Figure 5). In summer, $NO_2 + OH$ pathway showed a significantly higher

415 contribution during the daytime (55.1%) compared to nighttime (44.9%), which is attributed to

416 increased OH radical synthesis during longer days and higher temperatures in Lhasa (Rohrer and

417 Berresheim, 2006). A previous study indicated that lower $NO_2$ and higher $O_3$ concentrations enhance

418 the relative contribution of OH pathway to $NO_3^-$ formation (Wang et al., 2019). Additionally, the

419 concentration of ALWC (the detailed information is given in Text S3) was higher at night than during

420 the day in summer, favoring $NO_3^-$ formation through nocturnal formation. In winter, $f_{NO2+OH}$, $f_{NO3+VOC}$

421 and $f_{N2O5+H2O}$ were similar during both day and night. Typically, photolytic destruction and chemical

422 reactions with NO are rapid sinks during the daytime, with lifetimes generally less than 5 seconds and

423 resulting in extremely low concentrations. Similarly, the atmospheric lifetime of $N_2O_5$ under sunlight

424 is also very short (Wang et al., 2018). Thus, daytime $NO_3$ and $N_2O_5$ chemistry is often considered

425 negligible. However, a recent study revealed that a non-negligible amount of $NO_3$ radicals can persist

426 during the daytime in cold months, owing to the limited solar radiation (Hellén et al., 2018). Wang et

427 al. (2020a) found that the daytime production rate of $NO_3$ can be substantial due to elevated

428 concentrations of $O_3$ and $NO_2$, suggesting that the mixing ratios of $NO_3$ and $N_2O_5$ during the day may

429 not be negligible. Furthermore, in winter, lower temperatures and elevated $NO_2$ concentrations

430 facilitate a quasi-steady-state equilibrium between $NO_3$ and $N_2O_5$, slowing the overall reactivity of the

431 $NO_3^-$ precursors (Brown et al., 2003). This equilibrium condition minimizes diurnal fluctuations in

432 precursor concentrations, resulting in relatively stable nocturnal and daytime $NO_3^-$ formation pathways,

433 including $NO_3 + VOC$ and $N_2O_5 + H_2O$. Nevertheless, we acknowledge that the exact role of daytime

434 $NO_3/N_2O_5$ chemistry remains uncertain in Lhasa and should be further assessed using concurrent filed

435 observations or chemical transport models. Moreover, when interpreting the diurnal differences in

436 $\Delta^{17}O\text{-}NO_3^-$ values, the atmospheric lifetime of $NO_3^-$ must be considered. Given the atmospheric

437 lifetime of $NO_3^-$ is generally more than 12 hours, each sample might reflect both daytime and nighttime

$NO_3^-$ production impacting on $\Delta^{17}O$-$NO_3^-$ values (Park et al., 2004; Vicars et al., 2013).


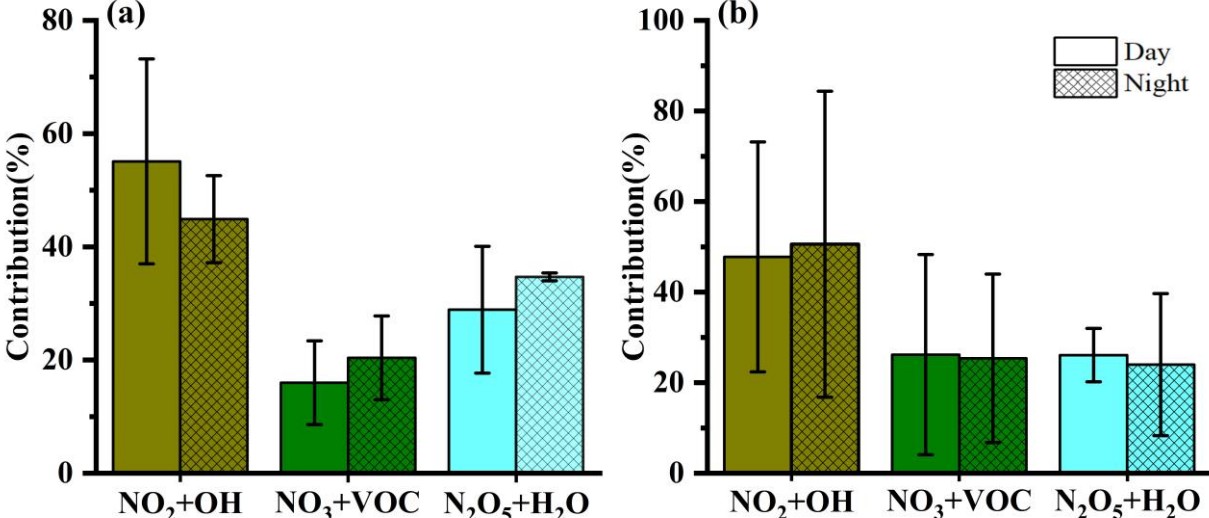


*Figure 5. the relative contributions (mean ± SD values) of $NO_2$ + OH, $NO_3$ + VOC, and $N_2O_5$ + $H_2O$*
*to $NO_3^-$ formation during the day and night (a) in summer and (b) winter in Lhasa during the sampling*
*campaign.*

**4.3 Integrated analysis of $NO_3^-$ oxidation pathways in Lhasa**

As shown in Figure S8, $NO_3$ + VOC pathway emerged as the major contributor to $NO_3^-$ formation
during periods of high $NO_3^-$ spikes. To elucidate the $NO_3^-$ formation pathways under different $NO_3^-$
concentrations, $NO_3^-$ samples were categorized into different concentration ranges (Figure 6). We
found the $f_{NO3+VOC}$ values increased and $f_{NO2+OH}$ values decreased with the $NO_3^-$ concentrations.
Although recent field radical measurements in urban sites in China found that OH and $HO_2$ radical
during haze period is comparable to clean days (Slater et al., 2020; Yang et al., 2021), our results
suggested that $NO_3$+VOC pathway still played an important role in $NO_3^-$ production under high-$NO_3^-$
concentration in Lhasa, possibly due to enhanced VOC emission. In addition to concentration effects,
meteorological factors typically also regulate the $NO_3^-$ oxidation pathways. Typically, high
temperature promotes the $NO_3^-$ formation in $f_{NO2+OH}$ values (Han et al., 2015). However, our study
revealed that the relationship between temperature and $f_{NO2+OH}$ values did not consistently show a
positive trend. Further analysis indicated that $NO_2$ and $O_3$ concentrations were negatively correlated,
with lower $NO_2$ concentrations paired with elevated $O_3$ levels (Figure S9). $f_{NO2+OH}$ values reached its
minimum when $NO_2$ was between 15 and 20 $\mu g/m^3$ and $O_3$ was within 100-120 $\mu g/m^3$. Although OH
radicals exhibit a higher oxidation potential (2.8 V) than $O_3$ (2.07 V), but atmospheric availability is
much lower than that of $O_3$ (Carslaw et al., 1999; Dubey et al., 1997). Therefore, $NO_2$ at lower
concentrations is more likely to be oxidized by OH than by $O_3$, even though $O_3$ concentrations were
high. With increasing $NO_2$ concentrations, the availability of OH radicals for oxidating $NO_2$ became
lower, resulting in a relatively higher proportion of $NO_2$ being oxidized by $O_3$ although $O_3$
concentrations were low. However, when the concentration of $O_3$ is below 20 $\mu g/m^3$, $O_3$ concentrations
were not sufficient to oxidize $NO_2$ due to the higher $NO_2$ concentrations and OH radicals for oxidating
$NO_2$ would re-dominate. These observations underscore that in high-altitude urban environments like
Lhasa, OH effectiveness is more important on $NO_3^-$ oxidation pathways than that of $O_3$. Additionally,
we identified an intriguing positive correlation between the atmospheric oxidizing capacity ($O_x = NO_2$
$+ O_3$) and $f_{NO3+VOC}$ values. $f_{NO3+VOC}$ values were lowest when $O_x$ was less than 90 $\mu g/m^3$, corresponding
to a maximum contribution from the $NO_2 + OH$ pathway. This suggests that $O_x$ is more indicative of
the pathways of $NO_3^-$ formation in the atmosphere compared to either $NO_2$ or $O_3$ alone. Typically,
High RH and ALWC were also positively correlated with $f_{N2O5+H2O}$. But RH was associated with
variable contributions from the $N_2O_5+H_2O$ pathway in our study, while increasing ALWC significantly
enhanced this pathway, indicating ALWC as a more reliable indicator of $NO_3^-$ formation.

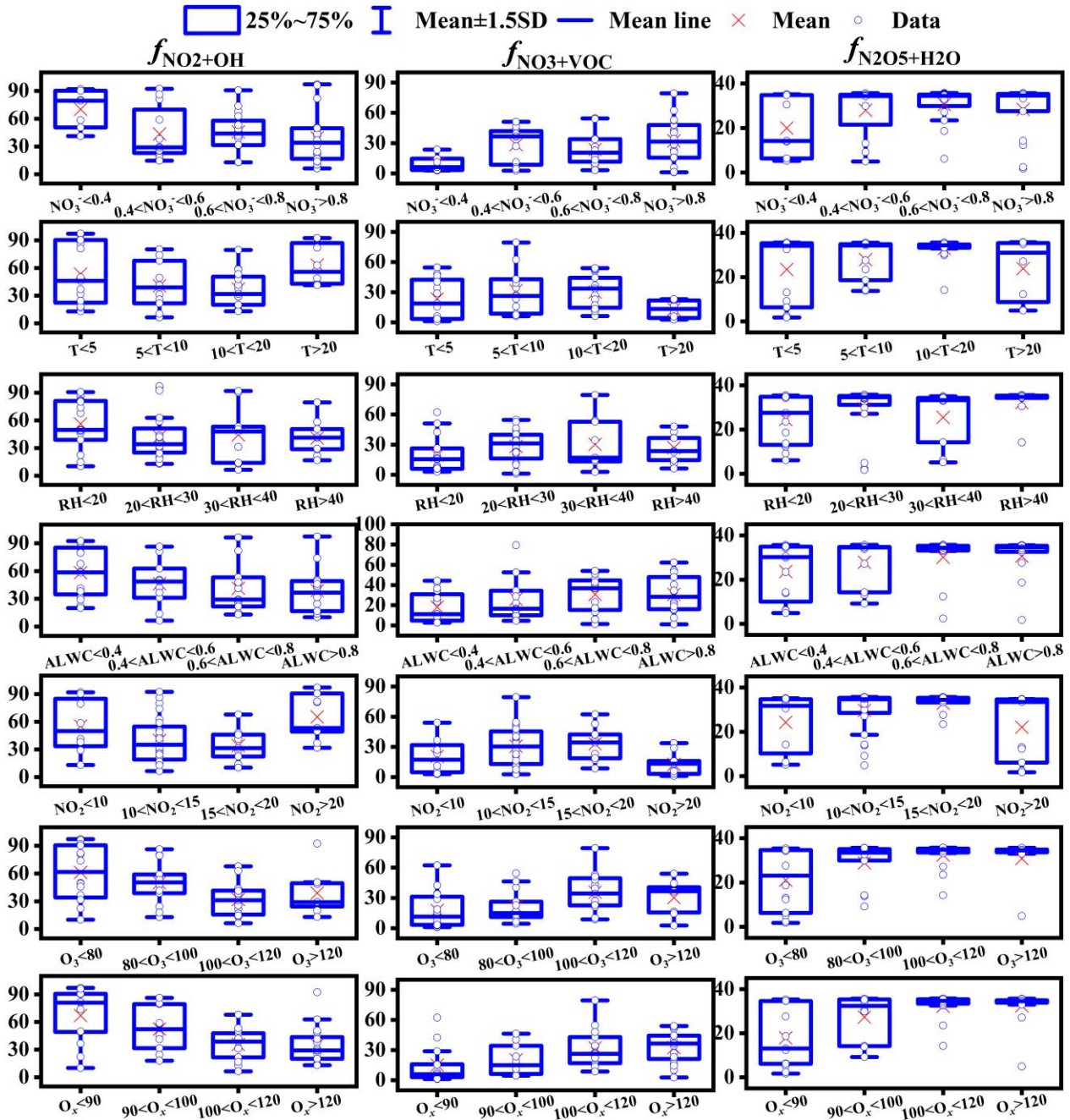


*Figure 6 Influence of $NO_3^-$ ($\mu g/m^3$), temperature (℃), RH (%), ALWC ($\mu g/m^3$), $NO_2$ ($\mu g/m^3$), $O_3$ and*

*$O_x$($\mu g/m^3$) on $NO_3^-$ formation pathways (%).*

## 4.4 Implications

The oxidation pathways of $NO_3^-$ in Lhasa, China, were constrained using a full year of $\Delta^{17}O$-$NO_3^-$ measurements from 2022 to 2023. Based on seasonal data, we observed a significant increase in the relative contribution of the $NO_3$+VOC to $NO_3^-$ formation during spring. Furthermore, the diurnal distribution of $NO_3^-$ oxidation pathways varied distinctly across seasons. To better understand the factors influencing these pathways, we integrated meteorological conditions, $NOx$ precursors, and

ALWC for a more comprehensive analysis of $NO_3^-$ formation. The results revealed that $Ox$ and ALWC are more reliable indicators of $NO_3^-$ oxidation pathways than meteorological factors. Notably, Lhasa's unique high-altitude environment such as strong solar radiation, persistently high $O_3$, and elevated VOC, promotes active $NO_3$ + VOC chemistry, especially in spring. Atmospheric ALWC is primarily produced by hygroscopic aerosols such as $SO_4^{2-}$, $NH_4^+$, and $Cl^-$. Therefore, in addition to controlling $NO_2$, $O_3$, and VOC, reducing these hygroscopic aerosols is crucial for effective $PM_{2.5}$ pollution control.

Although this study provides valuable insights into $NO_3^-$ formation mechanisms in Lhasa, we must acknowledge the associated uncertainties due to the lack of comprehensive observational constraints in Lhasa. Specifically, the limited understanding of local $RO_2$ concentrations led us to adopt empirical parameterizations and refer to measurements from other regions, which inevitably introduce uncertainty into the pathway apportionment. In addition, the absence of direct observations of nighttime NO emissions and the $NO_2$-NO isotope exchange processes in this region further complicates the interpretation of diurnal variations in $NO_3^-$ formation pathways. To improve the robustness of $\Delta^{17}O$-based pathway analysis, future studies should consider synchronous measurements of both $NO_2$ and $NO_3^-$ isotopes.

**Data availability**

All data are presented in the main text and/ or the Supplement. For additional data, please contact the corresponding author (liu.junwen@jnu.edu.cn).

**Author contributors**

JL designed, conceived, and led the research. XZ performed the data analysis and drafted the manuscript. JL, XZ NC and BB planned and carried out the measurements. NC, BB and PD were responsible for measuring the meteorological parameters. JL and PY secured funding for the continuous aerosol sampling and analysis. FC and YZ provided expertise on isotope analysis methods. JL offered guidance on data analysis, and all authors contributed to revising the manuscript.

**Competing interests**

The authors declare no competing financial interest.

**Acknowledgments**

This study was supported by the Natural Science Foundation of Xizang Autonomous Region (XZ202401ZR0067), Guangdong Basic and Applied Basic Research Foundation (2024B1515040026),

the second Tibetan Plateau Scientific Expedition and Research Program (20190ZKK0604) and Guangdong Provincial General Colleges and Universities Innovation Team Project (Natural Science) (2024KCXTD004).

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
