# Peer review of "The Critical Role of Volatile Organic Compounds Emission in Nitrate"

_EGUsphere, 2025_

## Author Comment (AC1)

Response to Anonymous Referee #1

*Summary: This manuscript presents an interesting study of aerosol nitrate formation in an elevated urban environment (Lhasa, Tibetan Plateau, China), with a focus on stable triple oxygen isotope measurements of aerosol nitrate. The authors collected high-volume aerosol samples for offline chemical and isotopic analyses and used the oxygen isotopic composition of nitrate ($\Delta^{17}O$) to infer the seasonality, and to a limited extent, diurnal variation of $NO_x$ oxidation and nitrate formation pathways. They conclude that oxidation by $NO_3$ + VOC contributes significantly to aerosol nitrate formation (~26%) based on a Bayesian isotope mixing model. While this study is timely and potentially impactful, particularly due to the high-elevation urban setting and the application of oxygen isotopes, it suffers from major methodological limitations and interpretive leaps. For example, the mixing model appears to be significantly underconstrained, and key assumptions (e.g., endmember $\Delta^{17}O$ values) are not adequately justified or tested. The manuscript would benefit from deeper contextualization, more rigorous uncertainty analysis, and supplemental modeling to support the stated conclusions. I believe the study has potential but requires substantial revision before it can be considered for publication in ACP.*

Response: Thanks for your valuable comments, which really helped improve the manuscript. Below, we will provide a detailed and point-by-point response to your comments. All the changes have been included in the latest manuscript (Reviewers' and Editorial Office's comments are in italics; our responses are in regular font).

**Comments:**

*1. Text S2: This section is never referenced in the main manuscript but contains a critical assumption about the fraction of $NO_2$ oxidation. Since the $\Delta^{17}O(NO_3^-)$ signature is largely derived from $NO_2$, this section should be moved to the main text.*

**Response:** Thanks for your suggestion. We have moved Text S2 to Section 2.4. **(Line 150-191)**

**Line 150-191: 2.4 Evaluation of NO$_3^-$ oxidation pathways**

In our study, we aimed to quantify the relative contribution of different oxidation pathways to $NO_3^-$ production based on $\Delta^{17}O\text{-}NO_3^-$. Due to the low $Cl^-$ concentrations observed in Lhasa, the $NO_3^-$ formation pathways considered in this study are limited to $NO_2+OH$, $NO_3+VOC$, and $N_2O_5+H_2O$. Although $NO_3+VOC$ is generally considered a minor pathway in continental regions (Alexander et al., 2009), we included it because elevated VOC concentrations were observed at our sampling site in Lhasa, influenced by both biogenic emissions (e.g. incense burning) and anthropogenic sources (e.g. vehicle emissions) (Tang et al., 2022). The relative contributions of the three pathways were determined using a $\Delta^{17}O$-based mass balance approach (Michalski et al., 2003), as shown in Equations (1) and (2):

$$\Delta^{17}O\text{-}NO_3^- = (\Delta^{17}O\text{-}NO_3^-)_{NO2+OH} \times f_{NO2+OH} + (\Delta^{17}O\text{-}NO_3^-)_{NO3+VOC} \times f_{NO3+VOC}$$
$$+(\Delta^{17}O\text{-}NO_3^-)_{N2O5+H2O} \times f_{N2O5+H2O} \quad (1)$$

$$f_{NO2+OH} + f_{NO3+VOC} + f_{N2O5+H2O} = 1 \quad (2)$$

where $\Delta^{17}O\text{-}NO_3^-$ value is the $\Delta^{17}O$ value of $NO_3^-$ in $PM_{2.5}$. The $(\Delta^{17}O\text{-}NO_3^-)_{NO2+OH}$, $(\Delta^{17}O\text{-}NO_3^-)_{NO3+VOC}$, and $(\Delta^{17}O\text{-}NO_3^-)_{N2O5+H2O}$ correspond to the $\Delta^{17}O$ values from $NO_2 + OH$, $NO_3 + VOC$ and $N_2O_5 + H_2O$, respectively. The $\Delta^{17}O$ values for each pathway were calculated using Equations (3), (4), and (5) (Savarino et al., 2016; Alexander et al., 2009):

$$(\Delta^{17}O\text{-}NO_3^-)_{NO2+OH} \ (‰) = 2/3\alpha \times \Delta^{17}O\text{-}O_3^* \quad (3)$$

$$(\Delta^{17}O\text{-}NO_3^-)_{NO3+VOC} \ (‰) = 2/3\alpha \times \Delta^{17}O\text{-}O_3^* + 1/3 \times \Delta^{17}O\text{-}O_3^* \quad (4)$$

$$(\Delta^{17}O\text{-}NO_3^-)_{N2O5+H2O} \ (‰) = 1/3\alpha \times \Delta^{17}O\text{-}O_3^* + 1/2(2/3\alpha \times \Delta^{17}O\text{-}O_3^* + 1/3 \times \Delta^{17}O\text{-}O_3^*) \quad (5)$$

Previous studies have demonstrated a linear correlation between $\Delta^{17}O\text{-}O_3$ and $\Delta^{17}O\text{-}O_3^*$, with $\Delta^{17}O(O_3)$ values ranging from 20% to 40% in tropospheric $O_3$ (Vicars and Savarino, 2014; Ishino et al., 2017). The equations are shown as follows (Vicars et al., 2012):

$$\Delta^{17}O\text{-}O_3^* = 1.5 \times \Delta^{17}O\text{-}O_3 \quad (6)$$

Based on previous observations of tropospheric $O_3$, $\Delta^{17}O\text{-}O_3^*$ average value was approximately 39‰. The $\alpha$ value represents the proportional contribution of $O_3$ to the NO oxidation pathway and can be estimated using the following equations (7)

(Alexander et al., 2009). When NO$x$ is in photochemical steady state, $\Delta^{17}$O-NO$_2$ can be represented using the following equation (10):

$$\alpha = K_{P1}[O_3] \times [NO]/(K_{P1} \times [O_3] \times [NO] + K_{P2} \times [NO] \times [HO_2] + K_{P3} \times [NO] \times [RO_2])$$
(7)

$$K_{P1} = 3.0 \times 10^{-12} \times e^{(-1500/T)} \text{ (8)}$$

$$K_{P2} = K_{P3} = 3.5 \times 10^{-12} \times e^{(270/T)} (cm^3 \cdot molecule^{-1} \cdot s^{-1}) \text{ (9)}$$

$$\Delta^{17}O\text{-}NO_2 = \alpha\Delta^{17}O\text{-}O_3\text{* (10)}$$

where T represents the ambient temperature (K) (Kunasek et al., 2008). The HO$_2$ mixing ratios were estimated using empirical equations in the absence of direct HO$_2$ observations (Kanaya et al., 2007). Due to the lower temperatures in Lhasa during non-summer seasons, HO$_2$ concentrations were assessed using a formula derived from winter conditions.

Winter

$$[HO_2\cdot]/ppt = \exp(5.7747 \times 10^{-2} [O_3] \text{ (ppb)} - 1.7227) \text{ for daytime (11)}$$

$$[HO_2\cdot]/ppt = \exp(7.7234 \times 10^{-2} [O_3] \text{ (ppb)} - 1.6363) \text{ for nighttime (12)}$$

Summer

$$[HO_2\cdot]/pptv = \exp(2.0706 \times 10^{-2} [O_3] \text{ (ppb)} + 1.0625) \text{ for daytime (13)}$$

$$[HO_2\cdot]/pptv = 0.2456 + 0.1841 [O_3] \text{ (ppb) for nighttime (14)}$$

*2. TOC Figure: The figure may mislead readers by implying significant seasonal differences in $\Delta^{17}O(NO_3^-)$ that are not statistically supported in the results. Differences were observed only in spring. Uncertainties must be included for the pathway contributions, and once added, the seasonal distinctions may not hold.*

**Response:** Thanks for your suggestion. We have revised the TOC figure to more accurately reflect the findings in the revised manuscript.

[Figure]

**3. Lines 35–37: The uncertainty in the calculated pathway contributions should be provided.**

**Response:** Thanks for your valuable suggestion. We have added it to the revised manuscript. **(Line 42-44)**

**Line 42-44:** Our results show that $NO_2 + OH$ is the largest contributor to $NO_3^-$ formation ($46 \pm 26\%$), followed by $NO_3 + VOC$ ($26 \pm 18\%$), and $N_2O_5 + H_2O$ ($28 \pm 11\%$) using the Bayesian Isotope Mixture Model.

**4. Lines 37–39: Explain how this difference between seasons was determined**

**Response:** Thanks for your suggestion. We have added it to the revised manuscript. **(Line 44-45)**

**Line 44-45:** Notably, there are significant differences in the $NO_2 + OH$, $NO_3 + VOC$, and $N_2O_5 + H_2O$ pathways between spring and other three seasons (T test, $p < 0.05$).

**5. Lines 39–40: Add context to how these statements were concluded.**

**Response:** Thanks for your suggestion. We have added it to the revised manuscript. **(Line 47-50)**

**Line 47-50:** By Hybrid Single-Particle Lagrangian Integrated Trajectory (HYSPLIT) dispersion model, we highlighted the influence of VOC emissions from regions such as Afghanistan and northern India, which enhanced $NO_3^-$ concentrations in Lhasa during spring.

**6. Line 42: Acronyms such as ALWC, NO₃⁻, and VOC are not defined in the abstract and should be introduced.**

**Response:** Thanks for your suggestion, we have added all acronyms to the revised manuscript. **(Line 39-41/Line 50-53)**

**Line 39-41:** Atmospheric particulate nitrate aerosol ($NO_3^-$), produced via the oxidation of nitrogen oxides ($NOx = NO + NO_2$), plays an important role in atmospheric chemistry and air quality, yet its formation mechanism remains poorly constrained in the plateau region.

**Line 50-53:** Furthermore, the diurnal distribution of $NO_3^-$ oxidation pathways varied distinctly across seasons, suggesting that these differences in $NO_3^-$ pathways are attributed to aerosol liquid water content (ALWC), volatile organic compounds (VOC) concentration, and pollution levels.

**7. Lines 87–89: It is unusual to target three "major" NO₃⁻ formation pathways in a continental setting, especially including NO₃ + VOC, which is generally minor. Consider referencing Alexander et al., 2019 and revisiting the classification of major pathways.**

**Response:** Thanks for your suggestion. (1) We acknowledge that in many continental environments, $NO_3 + VOC$ pathway is generally considered a minor contributor to $NO_3^-$ formation. However, recent observation-based studies have increasingly reported that this pathway can play a significant role in $NO_3^-$ production under certain atmospheric conditions (Zhang et al., 2022; Fan et al., 2021; Feng et al., 2023; Li et al., 2022).

(2) Our sampling site is located in the central urban area of Lhasa, where air masses are influenced by both biogenic and anthropogenic sources of Volatile Organic Compounds (VOC), including biomass burning and incense burning. Tang et al. (2022) have shown that VOC concentrations in Lhasa are comparable to those in the North China Plain, supporting the plausibility of a significant contribution from the $NO_3 + VOC$ pathway in this region.

(3) To address the reviewer's concern, we have now cited the perspective provided by

Alexander et al. (2009) in the revised manuscript. We have also clarified that the classification of the major pathways in this study is based on region-specific observational evidence, rather than general assumptions **(Line 156-159)**.

**Line 156-159:** Although $NO_3$ + VOC was generally considered a minor pathway in continental regions (Alexander et al., 2009), we included it because elevated VOC concentrations were observed at our sampling site in Lhasa, influenced by both biogenic emissions (e.g. incense burning) and anthropogenic sources (e.g. vehicle emissions) (Tang et al., 2022).

**8. *Lines 93–94: Since the site is at high elevation, provide information on altitude-related meteorology and its influence on boundary layer mixing and transport.***

**Response:** Thanks for your suggestion. We have added it to the revised manuscript. **(Line 111-114)**

**Line 111-114:** The strong solar radiation and large diurnal temperature variations in this sampling site can lead to pronounced changes in boundary layer height, which in turn significantly influence vertical mixing and the transport of air pollutants.

**9. *Lines 94–95: Include more detail on the urban characteristics and land use of Lhasa to contextualize emissions.***

**Response:** Thanks for your suggestion. We have added it to the revised manuscript. **(Line 107-111)**

**Line 107-111:** $PM_{2.5}$ samples were collected on the roof of a building (~15 m above ground) at the Meteorological Bureau of Lhasa (91.08°E, 29.40°N; Figure 1) in China. Lhasa, the capital of the Tibet Autonomous Region, is a rapidly developing city with a population of ~ 950000 and an urban area of ~ 30000 $km^2$ (Lhasa). The sampling site is surrounded by mixed land use, including residential areas, government offices, religious temples and commercial zones, with minimal heavy industry.

**10. *Lines 125–126: The link did not work. Ensure that all supplemental data used in the manuscript is archived and accessible via a reliable digital repository.***

**Response:** Thanks for checking carefully. The link has been updated in our revised manuscript. **(Line 149-151)**

**Line 149-151:** Additionally, $NO_2$ and $O_3$ during the sampling campaign were downloaded from the National Meteorological Information Center (https://air.cnemc.cn:18007/).
* * *
*11. Lines 131–132: The isotope mixing model assumes known $\Delta^{17}O$ endmembers. How were these determined, particularly for $\Delta^{17}O(NO_2)$? Please explain the derivation or source of these values.*

**Response:** Thank you for your suggestion. We have added detailed explanation of the derivation of $\Delta^{17}O$ endmember values, particularly for $\Delta^{17}O(NO_2)$, in Section 2.4 of the revised manuscript.
* * *
*12. Lines 150–153: The MDL for $NO_3^-$ was given earlier (Line 114), but MDLs for other ions are missing here. Please include them.*

**Response:** Thanks for your comment. We have now included MDLs for other relevant ions in the revised manuscript. **(Line 136-138)**

**Line 136-138:** The method detection limits (MDLs) for $Cl^-$, $NO_3^-$, $SO_4^{2-}$, $Na^+$, $NH_4^+$, $K^+$, $Mg^{2+}$, and $Ca^{2+}$ were 0.001 mg/L, 0.001 mg/L, 0.003 mg/L, 0.02 mg/L, 0.01 mg/L, 0.02 mg/L, 0.006 mg/L, and 0.02 mg/L, respectively.
* * *
*13. Lines 154–157: The provided URL for the model is not a proper citation. Please cite the model formally and ensure access.*

Response: Thanks for checking carefully. We have replaced the proper URL in the manuscript. **(Line 225)**

**Line 225**: https://www.ready.noaa.gov/HYSPLIT.
* * *
*14. Lines 157–159: Add a supporting reference for the assumptions or parameterizations described here.*

Response: Thanks for your suggestion. We have added supporting references in the

manuscript. **(Line 225-227)**

**Line 225-227:** This model has been widely used for simulating the transport and dispersion trajectories of pollutants such as $PM_{2.5}$, VOC, $O_3$, and $NOx$, among others (He et al., 2022; Zhao et al., 2015; Cao et al., 2023).

*15. Lines 159–162: Why was 3,650 meters chosen for the model?*

Response: Thanks for your comment. The height of 3650 meters was chosen because it corresponds to the actual altitude of our sampling site in Lhasa. To ensure consistency between the modeled air mass trajectories and the observational data, we used the same elevation as the receptor height in the HYSPLIT simulations.

*16. Lines 173–175: The claim of an "opposite" trend is unclear; visually, it seems the trends are actually consistent. Please clarify.*

Response: Thank you very much for your careful review and insightful comment. We agree that the original expression "opposite trend" was inaccurate and could cause confusion. We have revised it in the revised manuscript. **(Line 241-243)**

**Line 241-243:** Solar radiation intensity exhibited a seasonal trend consistent to those of temperature and RH, peaking in summer (394 $W/m^2$) and reaching its lowest levels in winter (220 $W/m^2$).

*17. Lines 179–200: Nitrate concentrations depend strongly on gas-particle partitioning, which is influenced by chemical composition (e.g., $NH_4^+$) and meteorology. Discuss the observed spring peak in $NO_3^-$ alongside $NH_4^+$ trends and partitioning behavior of $HNO_3$.*

**Response:** Thanks for your valuable comment. Although we did not measure gaseous $HNO_3$ in this study, we conducted correlation analysis between $NO_3^-$ and $NH_4^+$, $Ca^{2+}$, and $K^+$ concentrations, as well as temperature and relative humidity. We acknowledge the absence of $HNO_3$ data limits a full assessment of partitioning behavior. We will address this aspect more comprehensively in future studies using online gas-phase measurements and thermodynamic modelling **(Line 252-258).**

**Line 252-258:** $NO_3^-$ mass concentrations ranged from 0.10 to 1.72 μg/m$^3$, with an average value of 0.62 ± 0.31 μg/m$^3$. $NO_3^-$ concentrations exhibited distinct seasonal patterns. As shown in Figure S1, the equivalent concentrations of $[SO_4^{2-} + NO_3^-]$ were considerably higher than those of $[NH_4^+]$, indicating that $NH_4^+$ was insufficient to fully neutralize $NO_3^-$. This suggests that a portion of $NO_3^-$ may have existed in other forms, such as $KNO_3$ and $Ca(NO_3)_2$. This inference is supported by the strong positive correlations between $NO_3^-$ and $K^+$ (r = 0.64, p < 0.1) and $Ca^{2+}$ (r = 0.43, p < 0.01), especially in spring, as shown in Figure S2. In contrast, $NO_3^-$ showed relatively weak negative correlations with T (r = -0.27, p < 0.01) and RH (r = -0.22, p < 0.1), indicating that under the specific atmospheric conditions in Lhasa, meteorological parameters might not be the dominant factors controlling the gas-particle partitioning of $NO_3^-$. The maximum monthly average values of $NO_3^-$ concentration occurred in spring (0.83 ± 0.35 μg/m$^3$) with the instantaneous maximum reaching 1.72 μg/m$^3$, whereas the lowest was recorded in autumn (0.23 ± 0.13 μg/m$^3$) with an instantaneous minimum of only 0.09 μg/m$^3$ (Table 1). The elevated $NO_3^-$ concentrations in spring could be attributed to biomass burning emitted from south and Southeast Asia (Figure S3/Figure S4). The strong between $NO_3^-$ and $K^+$ in spring further this explanation.

[Figure]

*Figure S1 Equivalent concentrations of SO₄²⁻+NO₃⁻/NH₄⁺ in Lhasa during the sampling campaign.*

[Figure]

*Figure S2 Relationships between $NO_3^-$ and other parameters. The relationship between $NO_3$ and (a) $NH_4^+$ concentrations, (b) $K^+$ concentrations (c) $Ca^{2+}$, (d) T, and (e) RH; The red and blue represent spring and other seasons.*

**18. Lines 186–188: Provide a possible explanation for the observed concentration increase. Is there a local emission or meteorological reason?**

Response: Thanks for your suggestion. We have added a more detailed explanation for the observed springtime $NO_3^-$ concentration increase. Specifically, we suggest that the weak southeasterly winds may have limited atmospheric dispersion, leading to local accumulation of $NO_3^-$ precursors. Furthermore, the southeast sector may be influenced by traffic or agricultural sources, which could contribute to the enhanced $NO_3^-$ production. We have added it to the revised manuscript. (**Line 265-270**)

**Line 265-270:** The southeasterly sector of sampling site includes residential areas, agriculture land and major transportation routes, which are potential NO$x$ sources. In spring, intensified agriculture activities (e.g., fertilization, biomass burning) increase emissions. Meanwhile, low wind speeds likely limit atmospheric dispersion, promoting the local accumulation of precursors and enhancing $NO_3^-$ production.

**19. Lines 190–191: The COVID-19 shutdown period seems to have ended before sampling. If not, describe local shutdown policies in the methods section.**

**Response:** Thanks for your careful and constructive comment. We confirm that part of the sampling period, specifically during autumn 2022, coincided with strict COVID-19 control measures in Lhasa. During this period, the city implemented targeted lockdowns with near-total restrictions on vehicle traffic and pedestrian movement. We have added a description of these local measures in the methods section in the revised manuscript to clarify the context during sampling **(Line 121-123)**.

**Line 121-123:** During the autumn of 2022, Lhasa experienced intermittent COVID-19 control measures, including restricted movement, reduced traffic activity, and temporary lockdowns in urban areas (Daily).

**20. Lines 191–195: This statement implies minimal local influence. Consider emphasizing regional transport instead.**

**Response:** Thanks for pointing this out. We acknowledge that the original statement may have underemphasized the potential role of regional transport. We have revised the relevant sentences to clarify that although local emissions were suppressed due to COVID-19 lockdowns in autumn, the persistence of detectable $NO_3^-$ concentrations under stagnant conditions suggests a likely contribution from regional transport. We have revised it in the manuscript **(Line 270-284)**.

**Line 270-284:** During the rainy summer, shorter $NO_3^-$ lifetimes indicated a weak influence from regional transport, with a more pronounced contribution from local emissions. In autumn, $NO_3^-$ concentrations were relatively low, which coincided with strict local COVID-19 restrictions in Lhasa. These measures significantly reduced human activity and traffic, leading to suppressed local emissions. Despite low wind speeds typically favor pollutant accumulation, $NO_3^-$ concentrations remained low, suggesting that both reduced local sources and seasonal meteorological conditions constrained $NO_3^-$ production. Nevertheless, the persistence of measurable $NO_3^-$ under such stagnant conditions also implied a potential contribution from regional transport during this period. In winter, elevated $NO_3^-$ concentrations under low wind speeds ($< 3$

m/s) emphasized the significant contribution of local emissions. These findings underscored that both regional transport and local emissions were important contributors to $NO_3^-$ concentrations in Lhasa.

**21. Lines 193–195: If COVID-19 restrictions impacted emissions, explain why no corresponding impact is evident in your data.**

**Response:** Thanks for insightful comment. Although local COVID-19 restrictions likely reduced emissions, $NO_3^-$ was still detectable during autumn, possibly due to background levels and regional transport. We have clarified this in the revised text by acknowledging that reduced local emissions alone may not fully explain the observed concentrations, and that regional transport may have contributed to the persistence of $NO_3^-$.

**22. Lines 213–215: The data do not clearly show seasonal differences. Spring appears elevated, but other seasons are similar. Please clarify the interpretation.**

Response: Thanks for your helpful comment. We agree with your assessment. In the revised manuscript, we have removed the original sentence that inaccurately suggested clear seasonal variation in $\Delta^{17}O\text{-}NO_3^-$ values. **(Line 298-300:)**

**Line 298-300**: As shown in Table S2, the observed $\Delta^{17}O\text{-}NO_3^-$ values in this study were similar to most mid- and low-latitude regions, but lower than those in polar regions (~ 32‰). As listed in Table S1, the average $\Delta^{17}O\text{-}NO_3^-$ values in spring, summer, autumn, and winter were 28.8 ± 8.0‰, 25.5 ± 2.20‰, 25.6 ± 1.35‰, and 25.9 ± 3.56‰, respectively.

**23. Lines 216–218: Why was $NO_2$ formation not included in the discussion? It's central to $\Delta^{17}O(NO_3^-)$.**

**Response:** Thanks for your suggestion. We acknowledge that $NO_2$ formation plays a central role in controlling the $\Delta^{17}O\text{-}NO_3^-$. While direct measurements of $NO_2$ were not available during the campaign, we have addressed the NO-$NO_2$ conversion process indirectly through the parameter α, which represents the relative contribution of $O_3$ to

NO oxidation (Section 4.1). We acknowledge, however, that the absence of direct $NO_2$ observations introduces uncertainty, and we will consider the inclusion of $NO_2$ measurements in future field campaigns to better constrain this process. **(Line 326-333)**

**Line 326-333:** Typically, observations of $\Delta^{17}O\text{-}NO_3^-$ and estimated α (the proportion of $O_3$ oxidation in $NO_2$ production rate) values are employed to quantify the contributions of major $NO_3^-$ oxidation pathway in conjunction with a Bayesian model. The α value ranged from 0.63 to 0.93, with an average of 0.83 ± 0.06, suggesting the significance of $O_3$ participation in NO oxidation during the sampling campaign. On the other hand, our α values were lower than those (0.85-1) for other midlatitude regions (Alexander et al., 2009). The α values are influenced by the relative amount of $O_3$, $HO_2$ and $RO_2$ in NO*x* cycling. Due to the generally high $O_3$ concentrations ($O_3 > 50$ ppb) observed in Lhasa, nearly all α values exceeded 0.8 (Figure S6).

***24. Lines 220–223: Consider including a plot of this data.***

**Response:** Thanks for your suggestion. We added it to the revised manuscript. **(Figure S5)/ (Line 307-308)**

[Figure]

*Figure S5 Diurnal variation of $\Delta^{17}O\text{-}NO_3^-$ values in summer and winter during the sampling campaign*

**Line 301-303:** In contrast, the lower $\Delta^{17}O\text{-}NO_3^-$ values in other three seasons suggested a greater production of $NO_3^-$ formation via $NO_2 + OH$ pathway, leading to more negative $\Delta^{17}O\text{-}NO_3^-$ values. Diurnal variation in $\Delta^{17}O\text{-}NO_3^-$ values also differed across season (Figure S5).

**25. *Lines 240–241: A more detailed description of how alpha was determined is needed. A supplementary figure showing alpha and estimated $\Delta^{17}O(NO_2)$ over time would strengthen this section.***

**Response:** Thanks for your suggestion. A more detailed explanation of how the α value was determined has now been added in Section 2.4 of the revised manuscript. In particular, we clarified that α was estimated based on the relative contributions of $O_3$, $HO_2$, and $RO_2$ to $NO_2$ production, using their respective concentrations and reaction rate constants during the observation period.

To strengthen this section, we have included a new supplementary figure (Figure S6) presenting the time series of $O_3$, $HO_2$, $RO_2$ and α. Regarding $\Delta^{17}O(NO_2)$, we acknowledge that we were not able to directly measure $\Delta^{17}O$ in ozone ($\Delta^{17}O\text{-}O_3$) due to instrumental and logistical constraints. Instead, we adopted a literature-based value of $\Delta^{17}O\text{-}O_3^* = 39‰$. Because the $\Delta^{17}O$ in $NO_2$ is calculated via the equation ($\Delta^{17}O\text{-}NO_2 = \alpha \times \Delta^{17}O\text{-}O_3^*$), the temporal trend of estimated $\Delta^{17}O\text{-}O_3^*$ is similar to the α. We have added this explanation to the text. We also note that future work will aim to directly measure $\Delta^{17}O\text{-}O_3$ under high-altitude conditions like Lhasa to improve the estimation of $\Delta^{17}O\text{-}NO_2$. **(Line 321-328)**.

**Line 321-328**: Typically, observations of $\Delta^{17}O\text{-}NO_3^-$ and estimated α (the proportion of $O_3$ oxidation in $NO_2$ production rate) values are employed to quantify the contributions of major $NO_3^-$ oxidation pathway in conjunction with a Bayesian model. The α value ranged from 0.63 to 0.93, with an average of 0.83 ± 0.06, suggesting the significance of $O_3$ participation in NO oxidation during the sampling campaign. On the other hand, our α values were lower than those (0.85-1) for other midlatitude regions (Alexander et al., 2009). The α values are influenced by the relative amount of $O_3$, $HO_2$ and $RO_2$ in NO*x* cycling. Due to the generally high $O_3$ concentrations ($O_3 > 50$ ppb) observed in

Lhasa, nearly all α values exceeded 0.8 (Figure S6).

[Figure]

*Figure S6 Time series of (1) α value; (2) HO₂ and RO₂ concentrations; (3) O₃ concentrations during the sampling campaign. The volume mixing ratios were calculated from mass concentrations (μg/m³) based on the local atmospheric pressure and temperature conditions in Lhasa.*

**26. Lines 244–246: The pathway model lacks independent validation. Aside from the mixing model (which appears underconstrained), consider constructing a simple box model to test the plausibility of the proposed NO₃⁻ formation routes.**

**Response:** Thanks for your thoughtful suggestion. We fully agree that independent validation using a mechanistic model, such as a box model, would help further test the plausibility of the proposed NO₃⁻ formation pathways. However, our current dataset is based on a year-long field observation, which is robust in terms of temporal coverage but lacks the necessary time-resolved gas-phase precursors (e.g., NO₃, N₂O₅, VOC) required to reliably constrain a box model. Therefore, we were unable to conduct an independent model validation at this stage. We acknowledge this as a limitation of the current study, and we have added a note (**4.4 Implication**) in the revised manuscript to highlight this point. In future work, we plan to incorporate box modeling and/or online measurements of key reactive species to better constrain the chemical mechanisms

involved.

**27. Lines 252–253: In most continental urban settings, NO₃ + VOC is a minor contributor to aerosol nitrate. Reassess this conclusion in light of existing literature**

**Response:** Thank you for your suggestion. We acknowledge that in many continental urban environments, the $NO_3$ + VOC pathway is typically considered a minor contributor to aerosol nitrate. However, Lhasa presents a distinct atmospheric setting, characterized by relatively high $O_3$ levels and significant VOC emissions from both local (e.g., biomass burning, incense burning, residential heating) and regional (e.g., long-range transport from South Asia) sources, especially during spring.

Growing evidence suggests that the southern Tibetan Plateau, including Lhasa, is impacted by long-range transported pollutants from South Asia in spring. As a result, it is likely that both locally emitted and transported VOC could participate in nocturnal $NO_3$ chemistry via the $NO_3$ + VOC pathway. Nonetheless, we acknowledge the need for further observational and modelling studies to more quantitatively assess the importance of this pathway under high-altitude and complex emission conditions.

**28. Lines 254–256: If VOC data are available, use them to estimate the contribution of the NO₃ + VOC pathway. Also, at this elevation, stratospheric intrusions may occur. Could this be a source of high Δ¹⁷O nitrate?**

**Response: Thanks for your insightful comment.**

**(1) Regarding the use of VOC data to estimate the contribution of the NO₃ + VOC pathway:**

Unfortunately, VOC concentrations were not simultaneously measured during our sampling campaign. Although Tang et al. (2022) reported that elevated VOC levels in Lhasa, the available VOC data had limited temporal resolution and lacked comprehensive speciation. As a result, it was not feasible to quantitatively constrain the $NO_3$ + VOC pathway using observational VOC data. Future work integrating VOC data into a kinetic box model or MCM framework will help improve constraints on this pathway.

**(2) Regarding the potential influence of stratospheric intrusion on $\Delta^{17}O\text{-}NO_3^-$:**

We acknowledge the possibility of stratospheric intrusions at high elevations, which may introduce ozone with elevated $\Delta^{17}O\text{-}NO_3^-$ values. Previous studies have reported stratospheric intrusion events at high-altitude sites such as Nepal (5079 m a.s.l.) and Qomolangma Station (4300 m a.s.l.) during spring and winter. Given Lhasa's elevation and geographic setting, similar events may occur and could contribute to the enhanced $\Delta^{17}O\text{-}NO_3^-$ values observed in spring. This possible influence of a mixed stratospheric-tropospheric $O_3$ has been noted as a factor in $NO_3^-$ formation during this period. We have added it to the revised manuscript. **(Line 410-419)**

**Line 410-419:** A significant increase in the $f_{NO3+VOC}$ values was observed in spring (p < 0.05). First, $O_3$ and $NO_2$ are precursors of $NO_3$. In this work, the highest concentrations of $O_3$ were found in spring (114.9 ± 18.1 μg/m$^3$), likely leading to elevated $NO_3$ concentrations. Additionally, the low temperature and reduced OH radical concentrations in spring facilitate the reaction of $NO_2$ and $O_3$ to synthesize $NO_3$. This might be an appropriate reason for the $f_{NO3+VOC}$ values in spring. High-altitude locations such as Nepal (5079 m a.s.l.) and Qomolangma Station (4300 m a.s.l.) have experienced stratospheric ozone intrusions, especially in spring and winter, as reported in previous studies (Zhang et al., 2025; Cristofanelli et al., 2010; Morin et al., 2007; Zhang et al., 2022; Lin et al., 2016; Yin et al., 2017; Wang et al., 2020b). Notably, such intrusions in spring may elevate tropospheric $O_3$ levels in Lhasa, resulting in a mixture of tropospheric and stratospheric $O_3$ that enhances $NO_3^-$ production.

*29. Lines 283–284: The logic in this sentence doesn't follow clearly from the preceding text. Please revise.*

Response: Thank you for pointing this out. We have revised the sentence to improve the logical flow. **(Line 398-403)**

**Line 393-402:** Figure S7 illustrates the seasonal variations in the relative contributions of the three main oxidation pathways to $NO_3^-$ formation. When comparing different seasons, the $f_{NO2+OH}$ values were lower (p < 0.01) in spring (22.6%) than in winter (50.8%), summer (52.9%) and autumn (73.2%). The dominance of $NO_2$ + OH pathway in autumn is consistent with observations at Mt. Everest during the autumn seasons of

2017 and 2018, suggesting that $NO_3^-$ formation on the Tibetan Plateau in autumn may be mainly driven by $NO_2$ + OH pathway (Lin et al., 2021; Wang et al., 2020b).

**30. Lines 286–287: High $O_3$ levels increase $\Delta^{17}O(NO_2)$, which strongly influences $\Delta^{17}O(NO_3^-)$. This should be acknowledged explicitly.**

**Response:** Thank you for the insightful comment. We agree that high $O_3$ levels can elevate $\Delta^{17}O(NO_2)$, which in turn strongly affects $\Delta^{17}O(NO_3^-)$. Although we did not directly measure $\Delta^{17}O$ of $NO_2$ in this study, we evaluated the impact of $O_3$ on $\Delta^{17}O(NO_3^-)$ through the parameter α, which reflects the relative importance of $NO_2$ oxidation pathways involving $O_3$. In Section 4.1 of the revised manuscript, we have explicitly discussed the influence of elevated $O_3$ concentrations on the α value in Lhasa.

**31. Lines 295–297: Why is VOC assumed to be the only contribution from the biomass burning plume? Could oxidized nitrogen compounds also be transported?**

**Response:** Thanks for your insightful comment. In our manuscript, VOC was considered a major contributor from the biomass burning primarily due to two reasons. First, previous studies have shown that VOC concentrations in Lhasa are comparable to those in the North China Plain, suggesting a relatively high local VOC level (e.g., Li et al., 2020). Second, our results indicate that the relative contribution of the $NO_3$ + VOC pathway is significantly elevated in spring, a season when long-range transport from South Asia is active. Previous studies have reported that VOC originating from biomass burning and industrial emissions in South Asia can be transported to the Tibetan Plateau, leading to increased VOC concentrations. We therefore infer that the enhanced $f_{NO3+VOC}$ in spring is largely driven by elevated VOC levels from both local and transported sources.

Nonetheless, we agree with the reviewer that oxidized nitrogen species (e.g., NO$x$,) are also present in biomass burning and may be co-transported to the region. These reactive nitrogen compounds could further participate in $NO_3^-$ formation and cannot be ruled out as contributors. We have added clarification in the revised manuscript. **(Line 410-440)**

**Line 410-440:** A significant increase in the $f_{NO3+VOC}$ values was observed in spring ($p < 0.05$). First, $O_3$ and $NO_2$ are precursors of $NO_3$. In this work, the highest concentrations of $O_3$ were found in spring ($114.9 \pm 18.1$ μg/m$^3$), likely leading to elevated $NO_3$ concentrations. Additionally, the low temperature and reduced OH radical concentrations in spring facilitate the reaction of $NO_2$ and $O_3$ to synthesize $NO_3$. This might be an appropriate reason for the $f_{NO3+VOC}$ values in spring. High-altitude locations such as Nepal (5079 m a.s.l.) and Qomolangma Station (4300 m a.s.l.) have experienced stratospheric ozone intrusions, especially in spring and winter, as reported in previous studies (Zhang et al., 2025; Cristofanelli et al., 2010; Morin et al., 2007; Zhang et al., 2022; Lin et al., 2016; Yin et al., 2017; Wang et al., 2020b). Notably, such intrusions in spring may elevate tropospheric $O_3$ levels in Lhasa, resulting in a mixture of tropospheric and stratospheric $O_3$ that enhances $NO_3^-$ production. Second, previous study has indicated that the Afghanistan-Pakistan-Tajikistan region, the Indo-Gangetic Plain, and Meghalaya-Myanmar region could transport industrial VOC to various zones in Tibet from west to east. Additionally, agricultural areas in northern India could contribute biomass burning-related VOC to the middle-northern and eastern regions of Tibet (Li et al., 2017). During our sampling campaign, South and Southeast Asia air clusters were notably prevalent in the springtime, coinciding with intensive fire spots observed in Afghanistan, Pakistan, India, Nepal, and Bhutan (Figure S3/S4). These observations, combined with the prevailing South and Southeast Asia air mass trajectories in spring, strongly suggest that long-range transported VOC from South Asia were delivered to Lhasa and likely participated in local $NO_3^-$ production via $NO_3$ + VOC pathway. Moreover, recent studies have shown that ambient VOC concentrations in the urban areas on the Qinghai-Tibet Plateau were comparable to those in the North China Plain (Tang et al., 2022). The input of VOC through long-range transport might further elevate VOC concentrations, thereby promoting $NO_3^-$ formation via $NO_3$ + VOC pathway and contributing to the enhanced $f_{NO3+VOC}$ values observed in spring. While VOC appears to play a dominant role in the process, it should be noted that other nitrogen species (e.g., NO, $NO_2$) associated with biomass burning emissions may also be transported over long distances and influence $NO_3^-$ formation in

Lhasa. These co-transported nitrogen compounds, although not directly quantified in this study, could further contribute to $NO_3^-$ production in spring. Taken together, these findings provide strong evidence that long-range transport of biomass burning emissions, particularly from South Asia, can substantially influence springtime $NO_3^-$ formation in Lhasa.

**32. Lines 315–320: The diurnal nitrate interpretation doesn't account for the atmospheric lifetime of $NO_3^-$. Some residual $NO_3^-$ from nighttime may persist into the daytime. Please consider this in the discussion.**

**Response:** Thanks for your comment. In the revised manuscript, we have explicitly acknowledged the importance of the atmospheric lifetime of $NO_3^-$ when interpreting diurnal variations in $\Delta^{17}O$-$NO_3^-$. **(Line 480-484)**

**Line 480-484:** Moreover, when interpreting the diurnal differences in $\Delta^{17}O$-$NO_3^-$ values, the atmospheric lifetime of $NO_3^-$ must be considered. Given the atmospheric lifetime of $NO_3^-$ is generally more than 12 hours, each sample might reflect both daytime and nighttime $NO_3^-$ production impacting on $\Delta^{17}O$-$NO_3^-$ values (Park et al., 2004; Vicars et al., 2013).

**33. Lines 319–320: $NO_3$ and $N_2O_5$ chemistry is unlikely to significantly contribute to daytime $NO_3^-$ formation due to their short lifetimes in sunlight. Please calculate and discuss the expected lifetime.**

**Response:** Thanks for your comment. We fully agree that $NO_3$ and $N_2O_5$ radicals are highly reactive species with short atmospheric lifetimes under sunlight. Due to the limited availability of concurrent photolysis rate data and relevant concentrations ($NO_3$ and $N_2O_5$ concentrations) during our sampling campaign, we were unable to quantitatively calculate the daytime atmospheric lifetimes of these species at our site. Nevertheless, we have revised the manuscript to include a more explicit discussion based on previous studies. **(Line 450-484)**

**Line 450-484:** Interestingly, distinct diurnal patterns of $NO_3^-$ oxidation pathways were observed during the sampling campaign (Figure 5). In summer, $NO_2$ + OH pathway

showed a significantly higher contribution during the daytime (55.1%) compared to nighttime (44.9±%), which is attributed to increased OH radical synthesis during longer days and higher temperatures in Lhasa (Rohrer and Berresheim, 2006). A previous study indicated that lower $NO_2$ and higher $O_3$ concentrations enhance the relative contribution of OH pathway to $NO_3^-$ formation (Wang et al., 2019). Additionally, the concentration of ALWC (the detailed information is given in Text S3) was higher at night than during the day in summer, favoring $NO_3^-$ formation through nocturnal formation. In winter, $f_{NO2+OH}$, $f_{NO3+VOC}$ and $f_{N2O5+H2O}$ were similar during both day and night. Typically, photolytic destruction and chemical reactions with NO are rapid sinks during the daytime, with lifetimes generally less than 5 seconds and resulting in extremely low concentrations. Similarly, the atmospheric lifetime of $N_2O_5$ under sunlight is also very short (Wang et al., 2018). Thus, daytime $NO_3$ and $N_2O_5$ chemistry is often considered negligible. However, a recent study revealed that a non-negligible amount of $NO_3$ radicals can persist during the daytime in cold months, owing to the limited solar radiation (Hellén et al., 2018). Wang et al. (2020a) found that the daytime production rate of $NO_3$ can be substantial due to elevated concentrations of $O_3$ and $NO_2$, suggesting that the mixing ratios of $NO_3$ and $N_2O_5$ during the day may not be negligible. Furthermore, in winter, lower temperatures and elevated $NO_2$ concentrations facilitate a quasi-steady-state equilibrium between $NO_3$ and $N_2O_5$, slowing the overall reactivity of the $NO_3^-$ precursors (Brown et al., 2003). This equilibrium condition minimizes diurnal fluctuations in precursor concentrations, resulting in relatively stable nocturnal and daytime $NO_3^-$ formation pathways, including $NO_3$ + VOC and $N_2O_5$ + $H_2O$. Nevertheless, we acknowledge that the exact role of daytime $NO_3/N_2O_5$ chemistry remains uncertain in Lhasa and should be further assessed using concurrent filed observations or chemical transport models. Moreover, when interpreting the diurnal differences in $\Delta^{17}O$-$NO_3^-$ values, the atmospheric lifetime of $NO_3^-$ must be considered. Given the atmospheric lifetime of $NO_3^-$ is generally more than 12 hours, each sample might reflect both daytime and nighttime $NO_3^-$ production impacting on $\Delta^{17}O$-$NO_3^-$ values (Park et al., 2004; Vicars et al., 2013).

***35. Figure 5: Uncertainty/error bars are needed for all pathway contributions. These are model-derived estimates with inherent uncertainties and should not be presented as precise values.***

**Response:** Thanks for your suggestion. We agree that the pathway contributions are model-derived estimates and should be presented with appropriate uncertainties. In the revised manuscript, we have added error bars to represent the uncertainties associated with each pathway contribution.

[Figure]

*Figure 5. the relative contributions (mean ± SD values) of $NO_2$ + OH, $NO_3$ + VOC, and $N_2O_5$ + $H_2O$ to $NO_3^-$ formation during the day and night (a) in summer and (b) winter in Lhasa during the sampling campaign.*

***36. Lines 355–364: Much of the mechanistic discussion here is speculative. Consider using a simple model framework (e.g., kinetic or box model) to evaluate the chemical feasibility of the proposed pathways.***

**Response:** Thanks for your suggestion. We fully agree that incorporating a simple model framework such as a box or kinetic model would provide stronger mechanistic support for evaluating the chemical feasibility of the proposed pathways. However, due to the current lack of sufficient observational data (e.g., VOC, radical concentrations), we are unable to implement such a model in the present study. We acknowledge this limitation in the revised manuscript and will prioritize the development of a modelling component in our future work to improve the mechanistic understanding of $NO_3^-$ formation under high-altitude conditions.

[revised manuscript text omitted]

---

## Author Comment (AC2)

Response to Anonymous Referee #2

*Zheng et al., showed that volatile organic compounds play a critical role in the nitrate production on a plateau city, as inferred from the oxygen isotope anomaly of nitrate ($\Delta^{17}O\text{-}NO_3^-$). The $\Delta^{17}O\text{-}NO_3^-$ hold a wealth information about the atmospheric oxidation environment, which can be used to complement the model work as an observational constraint for NOx chemistry. I believe this study is of significant importance to the community as there are very sparse measurements of oxygen isotope anomaly of nitrate in high-elevation plateau environments. While I agree with most of the interpretation, some of the results, i.e., the day-night difference in $\Delta^{17}O\text{-}NO_3^-$ require further deliberation. In addition, considerable improvements could be made in the presentation of the results, refining the methodology, the layout of the figures, as well as enhancing the overall clarity of the writing. Overall, the manuscript should be subjected to major revisions listed below.*

**Response:** We would like to express our deepest gratitude for the constructure comments, which have significantly improved the quality of our work. Below, we provide detailed, point-by-point responses to all the reviewers' comments. All the changes have been included in the newest manuscript (Reviewers' and Editorial Office's comments are in italics; our responses are in regular font).

**Specific comments:**

*1: The author highlights that VOCs+$NO_3$ is of particular important for nitrate formation in Lhasa in spring based on the $\Delta^{17}O$ measurements and a simple mass-balance model calculation (i.e., Bayesian). The author did a lot of statistical analysis based on the Bayesian model outputs. It is well known that the Bayesian models of this nature was mathematically underdetermined and there was no unique solution with only one constraint but for three solutions (see Phillips et al., 2014), therefore model results will be associated with significant uncertainty. The comparison, statistical analysis and any conclusions draw from these results should be approached with great caution. For example, the contribution of OH+$NO_2$ likely*

***fluctuates around 50% throughout the year.***

**Response:** Thanks for your valuable suggestion. We fully acknowledge that Bayesian isotope mixing model (SIAR) used in our study is mathematically underdetermined when constrained by a single tracer ($\Delta^{17}$O-NO$_3^-$) and three potential pathways, as discussed in Phillips et al. (2014). As a result, the output carries inherent uncertainty, and the exact source apportionment solutions are not unique.

To assess the robustness of our findings, we conducted a sensitivity analysis by varying two key parameters: (1) α, the fraction of NO$_2$ oxidized by O$_3$, and (2) $\Delta^{17}$O of terminal O atoms in O$_3$ ($\Delta^{17}$O-O$_3$*). As shown in Table S1, both parameters influence the partitioning results. When $\Delta^{17}$O-O$_3$* was fixed at 39‰, increasing α from 0.7 to 0.9 led to a substantial increase in the contribution of the NO$_2$ + OH pathway (from 25% to 46%), while the NO$_3$ + VOC pathway decreased (from 46% to 25%). In contrast, the N$_2$O$_5$ + H$_2$O contribution remained nearly constant at 28-29%, indicating its relative insensitivity to α.

Similarly, when α was set between 0.8-0.9, increasing $\Delta^{17}$O-O$_3$* from 37‰ to 39‰ resulted in a rise in the NO$_2$ + OH contribution (from 37% to 46%) and a decline in NO$_3$ + VOC (from 35% to 26%), while the N$_2$O$_5$ + H$_2$O pathway again remained stable. These results suggest that although the NO$_2$ + OH and NO$_3$ + VOC pathways are sensitive to assumptions about α and $\Delta^{17}$O-O$_3$*, the N$_2$O$_5$ + H$_2$O pathway is relatively robust across the tested range.

Given that Lhasa is characterized by relatively high VOC concentrations and that $\Delta^{17}$O-O$_3$* is typically close to 39‰, we believe that our parameter assumptions are reasonable.

To address your concern, we have carefully revised the main text to avoid overstatements and now use more cautious language when discussing model-derived pathway contributions. **(Line 333-348)**

**Line 333-348:** To evaluate the impact of key parameters on the estimated contributions of different NO$_3^-$ formation pathways, we conducted a sensitivity analysis by assumed the α values and $\Delta^{17}$O value of the terminal oxygen atoms of O$_3$ ($\Delta^{17}$O-O$_3$*). As listed in Table S3, the assumption of α and $\Delta^{17}$O-O$_3$* have an impact on the NO$_3^-$ formation

mechanisms. When $\Delta^{17}O\text{-}O_3$* was fixed at 39‰, increasing $\alpha$ from 0.7 to 0.9 led to a notable increase in the relative contribution of the $NO_2$ + OH pathway from 25% to 46%, while that of the $NO_3$ + VOC pathway decreased from 46% to 25%. The $N_2O_5$ + $H_2O$ pathway remained nearly constant, with contributions ranging between 28% and 29%, indicating that this pathway is relatively insensitive to changes in $\alpha$ values. Similarly, when $\alpha$ was varied within a reasonable range (0.68-0.93), increasing the $\Delta^{17}O\text{-}O_3$* value from 37‰ to 39‰ led to an increase in the $NO_2$ + OH contribution from 37% to 46%, and a corresponding decrease in the $NO_3$ + VOC contribution from 35% to 26%. Again, the $N_2O_5$ + $H_2O$ contribution remained stable at ~ 28%. These results suggest that the estimated contributions of $NO_2$ + OH and $NO_3$ + VOC pathways are sensitive to assumptions about $\alpha$ and $\Delta^{17}O\text{-}O_3$*, whereas the contribution of the $N_2O_5$ + $H_2O$ pathway is relatively robust under the tested conditions. Because Lhasa is characterized by relatively high VOC concentrations and $\Delta^{17}O\text{-}O_3$* is generally close to 39‰ in the troposphere, we consider our parameter assumptions reasonable for further estimating $NO_3^-$ formation pathways for each sample.

Table S3 The average contribution of three oxidation pathways to $NO_3^-$ formation for the different the $\alpha$ values and $\Delta^{17}O\text{-}O_3$* values

| parameters | relative contribution of different oxidation pathways (%) | | |
|---|---|---|---|
| | NO₂+OH | NO₃+VOC | N₂O₅+H₂O |
| $\Delta^{17}O\text{-}O_3{}^*$=39‰ | | | |
| $\alpha$=0.7 | 25 | 47 | 28 |
| $\alpha$=0.8 | 41 | 29 | 30 |
| $\alpha$=0.9 | 46 | 26 | 28 |
| | | | |
| $\alpha$=0.68-0.93 | | | |
| $\Delta^{17}O\text{-}O_3$*=37‰ | 37 | 35 | 28 |
| $\Delta^{17}O\text{-}O_3$*=38‰ | 42 | 30 | 28 |
| $\Delta^{17}O\text{-}O_3$*=39‰ | 46 | 26 | 28 |

***2: Regarding the source of VOCs, the authors suggest that high ambient VOCs in spring may originate from South Asia via long-range transport. There are growing evidence that long-range transport of atmospheric pollutants from South Asia regulating the aerosol loadings in south of Tibetan Plateau in spring. Does nitrate aerosol in Lhasa also be impacted by the long-range transport, especially in spring? It is likely that the author assumed that long-range transported VOCs involve in the local nitrate production in Lhasa through $NO_3$+VOC pathways. This should be explicitly addressed in the main text.***

**Response:** Thanks for your insightful comment. As suggested, we have revised the main text to explicitly clarify the potential influence of long-range transported VOC on $NO_3^-$ formation in Lhasa, particularly during spring. We highlight that air masses originating from South Asia (e.g., Afghanistan, Pakistan, northern India, Nepal) during spring are likely to carry biomass burning-related VOC, which could enhance local $NO_3^-$ formation via the $NO_3$ + VOC pathway. We have also noted that elevated ambient VOC levels in urban areas on the Qinghai-Tibet Plateau, comparable to those in the North China Plain, may be further amplified by this long-range transport. Furthermore, we acknowledged the potential co-transport of nitrogenous species (e.g., NO, $NO_2$) associated with biomass burning that could also contribute to $NO_3^-$ formation. **(Line 410-440)**

**Line 410-440:** A significant increase in the $f_{NO3+VOC}$ values was observed in spring ($p < 0.05$). First, $O_3$ and $NO_2$ are precursors of $NO_3$. In this work, the highest concentrations of $O_3$ were found in spring (114.9 ± 18.1 μg/m$^3$), likely leading to elevated $NO_3$ concentrations. Additionally, the low temperature and reduced OH radical concentrations in spring facilitate the reaction of $NO_2$ and $O_3$ to synthesize $NO_3$. This might be an appropriate reason for the $f_{NO3+VOC}$ values in spring. High-altitude locations such as Nepal (5079 m a.s.l.) and Qomolangma Station (4300 m a.s.l.) have experienced stratospheric ozone intrusions, especially in spring and winter, as reported in previous studies (Zhang et al., 2025; Cristofanelli et al., 2010; Morin et al., 2007; Zhang et al., 2022; Lin et al., 2016; Yin et al., 2017; Wang et al., 2020b). Notably, such intrusions in spring may elevate tropospheric $O_3$ levels in Lhasa, resulting in a mixture

of tropospheric and stratospheric $O_3$ that enhances $NO_3^-$ production. Second, previous study has indicated that the Afghanistan-Pakistan-Tajikistan region, the Indo-Gangetic Plain, and Meghalaya-Myanmar region could transport industrial VOC to various zones in Tibet from west to east. Additionally, agricultural areas in northern India could contribute biomass burning-related VOC to the middle-northern and eastern regions of Tibet (Li et al., 2017). During our sampling campaign, South and Southeast Asia air clusters were notably prevalent in the springtime, coinciding with intensive fire spots observed in Afghanistan, Pakistan, India, Nepal, and Bhutan (Figure S3/S4). These observations, combined with the prevailing South and Southeast Asia air mass trajectories in spring, strongly suggest that long-range transported VOC from South Asia were delivered to Lhasa and likely participated in local $NO_3^-$ production via $NO_3$ + VOC pathway. Moreover, recent studies have shown that ambient VOC concentrations in the urban areas on the Qinghai-Tibet Plateau were comparable to those in the North China Plain (Tang et al., 2022). The input of VOC through long-range transport might further elevate VOC concentrations, thereby promoting $NO_3^-$ formation via $NO_3$ + VOC pathway and contributing to the enhanced $f_{NO3+VOC}$ values observed in spring. While VOC appears to play a dominant role in the process, it should be noted that other nitrogen species (e.g., NO, $NO_2$) associated with biomass burning emissions may also be transported over long distances and influence $NO_3^-$ formation in Lhasa. These co-transported nitrogen compounds, although not directly quantified in this study, could further contribute to $NO_3^-$ production in spring. Taken together, these findings provide strong evidence that long-range transport of biomass burning emissions, particularly from South Asia, can substantially influence springtime $NO_3^-$ formation in Lhasa.

*3: (1) The methodology for the determination of specific pathway contribution to nitrate based on $\Delta^{17}O$ should be clearly presented in the Method section. One of the most important of part is the A value (i.e, the relative importance of $O_3$ versus $RO_2$ in $NO_2$ formation), when using $\Delta^{17}O$ to distinguish nitrate formation pathways. First, I noticed that the author derived the $RO_2$ concentrations based on a empirical*

*relationship about $O_3$ mixing ratio. This relationship between $RO_2$ and $O_3$ indeed has been widely used in relevant study as concurrent $RO_2$ measurement is unavailable. This method is feasible at present. However, the relationship between $RO_2$ and $O_3$ the author used in this study is referred to Kanaya et al., 2007, which was conducted in urban site in central Tokyo. I believe that the atmospheric condition in Lhasa is completely different from that in Tokyo, i.e., the dominant $RO_2$ source. $RO_2$ production is majorly determined by solar radiations, which is also different between the two sites, as noticed in the Introduction. I recommend the calculation of $RO_2$ concentrations using MCM model and recent field observations of VOCs at Lhasa (see Chunxiang Ye et al., 2023).*

Response: Thanks for your valuable suggestion. We sincerely thank the reviewer for pointing out the importance of accurately estimating $RO_2$ concentrations in the calculation of the α value, which is indeed a critical parameter for $\Delta^{17}O$-based pathway apportionment. As noted, we adopted an empirical relationship between $RO_2$ and $O_3$ based on Kanaya et al. (2007), due to the absence of direct $RO_2$ observations and the lack of local model constraints. We fully acknowledge that this parameterization, developed for urban Tokyo, may not fully reflect the atmospheric conditions in high-altitude Lhasa, particularly given differences in radiation intensity and VOC composition.

Unfortunately, due to the lack of comprehensive VOC datasets and the necessary input parameters, we were unable to conduct a robust MCM (Master Chemical Mechanism) simulation for $RO_2$ in this study. We agree that such modelling, particularly with reference to recent VOC measurements in Lhasa, would significantly improve the accuracy of pathway estimation. We will prioritize this in our future work as more data becomes available.

*(2) Second, the author also suggests that nighttime $RO_2$ may play a role in the NOx oxidations. Similarly, the derivation of nighttime $RO_2$ is valid only when $O_3$ oxidation VOC dominates the $RO_2$ production (Kanaya et al., 2007). Nighttime $RO_2$ production mechanisms in Lhasa maybe unknown, however, in other urban cities such as*

*Beijing in China, NO$_3$ radical + VOC is the dominant channel for nighttime RO$_2$ production. In this case, nighttime RO$_2$ will be roughly correlated with the NO$_3$ radical production rate, $k_{O3+NO2[O3][NO2]}$. Although, given the high nighttime O$_3$ concentration in Lhasa, it maybe reasonable to assume O$_3$ dominant nighttime NO oxidation. To improve the robustness of the pathway differentiation, I recommend that this part could be done according to the approach of Alexander et al., 2020, and compare the field $\Delta^{17}O$-NO$_3^-$ measurements with the model results in Alexander et al., 2020.*

Response: Thanks for your insightful comment. Following your suggestion, we have further elaborated on the potential role of nighttime RO$_2$ in NO$_3^-$ formation in Lhasa and its possible link to NO$_3$ + VOC pathway. Although the exact mechanism of nighttime RO$_2$ production in Lhasa remains uncertain, previous studies have identified NO$_3$ + VOC reactions as the dominant source of RO$_2$ during nighttime. This process forms alkyl and multifunctional nitrates (RONO$_2$), which can undergo hydrolysis to yield HNO$_3$, contributing to NO$_3^-$ production. We also acknowledge the importance of O$_3$ and NO$_2$ in controlling the nighttime NO$_3$ radical production rate. Given that nighttime O$_3$ concentrations in Lhasa are relatively high, it is plausible that NO$_3$ radical levels are also elevated, enhancing NO$_3$ + VOC pathway. We have added it to the revised manuscript. **(Line 351-388)**

**Line 351-388:** On average, the relative contributions of NO$_2$ + OH ($f_{NO2+OH}$), NO$_3$ + VOC ($f_{NO3+VOC}$) and N$_2$O$_5$ + H$_2$O ($f_{N2O5+H2O}$) to NO$_3^-$ formation in Lhasa during the sampling campaign were 46 ± 26%, 26 ± 19% and 28 ± 11%, respectively. To better understand the characteristics of NO$_3^-$ formation mechanism in Lhasa, we performed a detailed comparison around the China for the relative contributions of key oxidation pathways using the $\Delta^{17}O$ methodology (Figure 4). Overall, similar to most Chinese cities, NO$_3^-$ formation in Lhasa was predominantly driven by the NO$_2$ + OH pathway, exhibiting distinct seasonal and regional variations. In particular, the average $f_{NO3+VOC}$ values were generally several times higher in spring in Lhasa than in other urban cities. Compared to rural/remote areas, the average $f_{NO3+VOC}$ values showed higher fractions in Lhasa, revealing the influence of anthropogenic emission, i.e., vehicle exhaust and

heating, on $NO_3^-$ formation. In Lhasa, the Capital of Tibet, field measurements among different years showed a substantial increase in VOC concentrations in urban areas of the Tibet Plateau, comparable to those in North China (Tang et al., 2022), revealing the importance of the active $NO_3$ + VOC pathway for $NO_3^-$ pollution formation in Lhasa. In fact, recent studies have recognized $NO_3$ + VOC as a major formation mechanism for $NO_3^-$ production. For instance, Fan et al. (2021) found that $f_{NO3+VOC}$ in Beijing increased from 17% in summer to 32% in winter based on $\Delta^{17}O$-$NO_3^-$ measurements. . He et al. (2018) estimated the relative contributions of $NO_3$ + VOC and $N_2O_5$ + $Cl^-$ to $NO_3^-$ formation and found that $NO_3$ + VOC and $N_2O_5$ + $Cl^-$ were in the range of 16-56%, underscoring the significant roles of these pathways during haze events in Beijing. Similarly, Feng et al. (2023) also reported that the $f_{NO3+VOC}$ values were up to 49.6% in winter in northern China. In Guangzhou,, Wang et al. (2023) noted that the average $f_{NO3+VOC}$ value was at the 488m (25%) higher than that at the ground (12%). Furthermore, Li et al. (2022) reported that $f_{NO3+VOC}$ increased from 5% in urban to 13.5% in rural regions in Northeast China. Although the specific nighttime $RO_2$ production mechanism in Lhasa remains unclear, studies in other cities have demonstrated that $NO_3$+VOC pathway was the dominant channel for nighttime $RO_2$ (Fisher et al., 2016), which in turn leads to the formation of alkyl and multifunctional nitrates ($RONO_2$) and eventually $NO_3^-$. In such cases, the $RO_2$ concentration is expected to be correlated with $NO_3$ radical production, which depends on the reaction rate of $O_3$ and $NO_2$ (Brown and Stutz, 2012). Given the relatively high nighttime $O_3$ concentrations in Lhasa, it is plausible that $O_3$-driven nighttime $NO_3$ chemistry plays an important role, thereby enhancing $NO_3$+VOC derived from $RO_2$ production and $NO_3^-$ formation. Global modelling studies also support the significant of this pathway. For instance, Alexander et al. (2020) reported that the $NO_3$ + VOC pathway via the $RONO_2$ mechanism accounted for 3% of global $NO_3^-$ formation on average. The relatively high $f_{NO3+VOC}$ values observed in Lhasa are broadly consistent with these findings, especially under conditions of high VOC concentrations and strong nighttime oxidant levels.

*4: I DONOT agree with the interpretation of the observed day-night differences in $\Delta^{17}O$-$NO_3^-$ during winter and summer (Lines: 307-333). Remember that daytime $NO_3$ and $N_2O_5$ chemistry should be negligible in nitrate chemistry, and no supporting evidence for this claim could be found in reference in Brown et al., 2011. Note high $NO_3$ production rate not means high mixing ratio of $NO_3$, $NO_3$ and $N_2O_5$ will be rapidly decomposed under sunlight. Although there are increasing studies showing the potential impact of daytime NO3 radical chemistry, the importance of daytime $NO_3/N_2O_5$ chemistry should be investigated with concurrent field observations or model experiments. The atmospheric residence time of nitrate should be considered for the comparison of day-night difference in $\Delta^{17}O$-$NO_3^-$, see Vicars et al., 2013.*

Response: Thanks for your valuable suggestion. Specifically, we acknowledge that under typical atmospheric conditions, $NO_3$ and $N_2O_5$ are rapidly photolyzed during the day, with reported $NO_3$ lifetimes of less than 10 seconds in sunlight, making their daytime accumulation negligible. However, we also note that several recent studies suggest that daytime $NO_3$ production rates can be non-negligible, especially in winter. To address the reviewer's concern, we have clarified the distinction between production rate and ambient concentration, and we now explicitly acknowledge the uncertainty in the role of daytime $NO_3/N_2O_5$ chemistry. Additionally, we now consider the atmospheric residence time of nitrate, as recommended (Vicars et al., 2013), which may lead to integrated contributions from both daytime and nighttime chemistry in each sample. **(Line 450-484)**

**Line 450-484:** Interestingly, distinct diurnal patterns of $NO_3^-$ oxidation pathways were observed during the sampling campaign (Figure 5). In summer, $NO_2$ + OH pathway showed a significantly higher contribution during the daytime (55.1%) compared to nighttime (44.9%), which is attributed to increased OH radical synthesis during longer days and higher temperatures in Lhasa (Rohrer and Berresheim, 2006). A previous study indicated that lower $NO_2$ and higher $O_3$ concentrations enhance the relative contribution of OH pathway to $NO_3^-$ formation (Wang et al., 2019). Additionally, the concentration of ALWC (the detailed information is given in Text S3) was higher at night than during the day in summer, favoring $NO_3^-$ formation through nocturnal

formation. In winter, $f_{NO2+OH}$, $f_{NO3+VOC}$ and $f_{N2O5+H2O}$ were similar during both day and night. Typically, photolytic destruction and chemical reactions with NO are rapid sinks during the daytime, with lifetimes generally less than 5 seconds and resulting in extremely low concentrations. Similarly, the atmospheric lifetime of $N_2O_5$ under sunlight is also very short (Wang et al., 2018). Thus, daytime $NO_3$ and $N_2O_5$ chemistry is often considered negligible. However, a recent study revealed that a non-negligible amount of $NO_3$ radicals can persist during the daytime in cold months, owing to the limited solar radiation (Hellén et al., 2018). Wang et al. (2020a) found that the daytime production rate of $NO_3$ can be substantial due to elevated concentrations of $O_3$ and $NO_2$, suggesting that the mixing ratios of $NO_3$ and $N_2O_5$ during the day may not be negligible. Furthermore, in winter, lower temperatures and elevated $NO_2$ concentrations facilitate a quasi-steady-state equilibrium between $NO_3$ and $N_2O_5$, slowing the overall reactivity of the $NO_3^-$ precursors (Brown et al., 2003). This equilibrium condition minimizes diurnal fluctuations in precursor concentrations, resulting in relatively stable nocturnal and daytime $NO_3^-$ formation pathways, including $NO_3$ + VOC and $N_2O_5$ + $H_2O$. Nevertheless, we acknowledge that the exact role of daytime $NO_3/N_2O_5$ chemistry remains uncertain in Lhasa and should be further assessed using concurrent filed observations or chemical transport models. Moreover, when interpreting the diurnal differences in $\Delta^{17}O$-$NO_3^-$ values, the atmospheric lifetime of $NO_3^-$ must be considered. Given the atmospheric lifetime of $NO_3^-$ is generally more than 12 hours, each sample might reflect both daytime and nighttime $NO_3^-$ production impacting on $\Delta^{17}O$-$NO_3^-$ values (Park et al., 2004; Vicars et al., 2013).

**General comment**

*1: The description of nitrate formation pathways (Text S1) and the associated $\Delta^{17}O$ signatures should be presented in the main text.*

Response: Thanks for your suggestion. We have added it to the main text in the revised manuscript (2.4).

*2: Line 62-63 Numerous field experiments have demonstrated that the $N_2O_5$ uptake probability on aerosol varied significantly, depending on the aerosol composition, meteorological parameters.*

Response: Thanks for your valuable suggestion. We have clarified this point in the revised manuscript by specifying that one of the major sources of uncertainty in modelling $NO_3^-$ formation via the $N_2O_5 + H_2O$ pathway is the wide variability in the $N_2O_5$ uptake probability ($\gamma$), which has been shown in numerous field experiments to depend strongly on aerosol composition and meteorological conditions such as temperature and relative humidity. **(Line 72-75)**

**Line 72-75:** However, there is considerable uncertainty in modelling the contribution of individual oxidation pathways to $NO_3^-$ formation, particularly for the $N_2O_5 + H_2O$ pathway, due to the wide variability of key parameters such as the $N_2O_5$ uptake coefficient, which has been shown to vary significantly with aerosol composition, relative humidity, and temperature.

*3: Line 237: I think the highlight of the text is the comparison of nitrate chemistry in high-elevation city with that in plain region. More discussion is needed to explore the mechanisms regulating the nitrate oxidation pathways, rather than a simple comparison of relative importance.*

Response: Thanks for your suggestion. Specifically, we have expanded the discussion to include potential mechanisms that may regulate the observed differences in oxidation pathways, with a particular focus on the $NO_3$ + VOC pathway. In this context, we discuss the role of elevated nighttime $O_3$ concentrations in Lhasa. Since this issue overlaps with the reviewer's Specific Comment 3 regarding the role of nighttime $RO_2$ and the importance of linking $NO_3$ radical production to ambient oxidant levels, we addressed both concerns together in the revised discussion to provide a more robust and coherent mechanistic interpretation.

*4: Line 345-347 Recent field radical measurements in urban sites in China found that OH and HO2 radical during haze period is comparable to clean days, see Slater*

*et al., 2020, Lu et al., 2019.*

**Response:** Thanks for your suggestion. We sincerely apologize for not citing the most up-to-date literature in our original manuscript. Initially, we referred to earlier studies, which may have led to an outdated understanding of radical behavior under polluted conditions. In response to the reviewer's helpful comment, we have carefully reviewed recent findings and incorporated updated references (Slater et al., 2020; Yang et al., 2021), which indicate that OH and $HO_2$ radical levels during haze periods can be comparable to those on clean days in urban China. We have revised the manuscript.

**(Line 502-511)**

**Line 502-511:** As shown in Figure S8, $NO_3$ + VOC pathway emerged as the major contributor to $NO_3^-$ formation during periods of high $NO_3^-$ spikes. To elucidate the $NO_3^-$ formation pathways under different $NO_3^-$ concentrations, $NO_3^-$ samples were categorized into different concentration ranges (Figure 6). We found the $f_{NO3+VOC}$ values increased and $f_{NO2+OH}$ values decreased with the $NO_3^-$ concentrations. Although recent field radical measurements in urban sites in China found that OH and $HO_2$ radical during haze period is comparable to clean days (Slater et al., 2020; Yang et al., 2021), our results suggested that $NO_3$+VOC pathway still played an important role in $NO_3^-$ production under high-$NO_3^-$ concentration in Lhasa, possibly due to enhanced VOC emission.

5: *Line 373 The implication sounds impotent. It is well known that aerosol liquid water content (ALWC) and Ox (oxidation capacity) regulate nitrate concentrations—ALWC impacts gas-to-particle partitioning, while Ox affects oxidation efficiency. The authors should focus on the specific or unique environmental conditions in the Tibetan Plateau that could be reflected by the measurements of $\Delta^{17}O$-$NO_3^-$.*

Response: Thanks for your suggestion. We have strengthened the Implications section to better highlight the unique environmental conditions of the Tibetan Plateau, particularly those that influence nitrate oxidation as captured by $\Delta^{17}O$-$NO_3^-$. Specifically, we now emphasize the role of high solar radiation, persistently elevated $O_3$ levels, and seasonally enhanced VOC concentrations in Lhasa, which together

promote active $NO_3$ + VOC chemistry — especially in spring. These features are characteristic of high-altitude urban environments and contribute to the distinct oxidation pathways observed in this region.

At the same time, we acknowledge the limitations of our study. Due to the lack of direct observations of $RO_2$ concentrations in Lhasa, we adopted empirical estimations based on other regions, which introduces uncertainty into the pathway apportionment. Additionally, the absence of measurements related to nighttime NO emissions and $NO_2$-NO isotope exchange in the region may affect the accuracy of the diurnal pattern interpretation. We have added these points to the revised Implications section to clarify the scope and robustness of our conclusions. **(Line 540-559)**

**Line 540-559:** The oxidation pathways of $NO_3^-$ in Lhasa, China, were constrained using a full year of $\Delta^{17}O$-$NO_3^-$ measurements from 2022 to 2023. Based on seasonal data, we observed a significant increase in the relative contribution of the $NO_3$+VOC to $NO_3^-$ formation during spring. Furthermore, the diurnal distribution of $NO_3^-$ oxidation pathways varied distinctly across seasons. To better understand the factors influencing these pathways, we integrated meteorological conditions, NO$x$ precursors, and ALWC for a more comprehensive analysis of $NO_3^-$ formation. The results revealed that O$x$ and ALWC are more reliable indicators of $NO_3^-$ oxidation pathways than meteorological factors. Notably, Lhasa's unique high-altitude environment such as strong solar radiation, persistently high $O_3$, and seasonally elevated VOC, promotes active $NO_3$ + VOC chemistry, especially in spring. Atmospheric ALWC is primarily produced by hygroscopic aerosols such as $SO_4^{2-}$, $NH_4^+$, and $Cl^-$. Therefore, in addition to controlling $NO_2$, $O_3$, and VOC, reducing these hygroscopic aerosols is crucial for effective $PM_{2.5}$ pollution control.

Although this study provides valuable insights into $NO_3^-$ formation mechanisms in Lhasa, we must acknowledge the associated uncertainties due to the lack of comprehensive observational constraints in Lhasa. Specifically, the limited understanding of local $RO_2$ concentrations led us to adopt empirical parameterizations and refer to measurements from other regions, which inevitably introduce uncertainty into the pathway apportionment. In addition, the absence of direct observations of

nighttime NO emissions and the $NO_2$-NO isotope exchange processes in this region further complicates the interpretation of diurnal variations in $NO_3^-$ formation pathways. To improve the robustness of $\Delta^{17}O$-based pathway analysis, future studies should consider synchronous measurements of both $NO_2$ and $NO_3^-$ isotopes.

*6: Additionally, many sentences throughout the manuscript require careful revision for clarity and grammar (e.g., Lines 31–33)*

Response: Thanks for your suggestion. We have carefully reviewed and revised the entire manuscript to improve the clarity, grammar, and overall readability of all sentences, including the example in Lines 31–33.

**Reference**

[revised manuscript text omitted]